# DYNAMIC KERNEL GRAPH SPARSIFIERS

## ABSTRACT

A geometric graph associated with a set of points $P = \{x_1, x_2, \cdots, x_n\} \subset \mathbb{R}^d$ and a fixed kernel function $\mathsf{K} : \mathbb{R}^d \times \mathbb{R}^d \to \mathbb{R}_{\geq 0}$ is a complete graph on $P$ such that the weight of edge $(x_i, x_j)$ is $\mathsf{K}(x_i, x_j)$. We present a fully-dynamic data structure that maintains a spectral sparsifier of a geometric graph under updates that change the locations of points in $P$ one at a time. The update time of our data structure is $n^{o(1)}$ with high probability, and the initialization time is $n^{1+o(1)}$. Under certain assumption, our data structure can be made robust against adaptive adversaries, which makes our sparsifier applicable in iterative optimization algorithms. We further show that the Laplacian matrices corresponding to geometric graphs admit a randomized sketch for maintaining matrix-vector multiplication and projection in $n^{o(1)}$ time, under *sparse* updates to the query vectors, or under modification of points in $P$.

## 1 INTRODUCTION

Kernel methods are a fundamental tool in modern data analysis and machine learning, with extensive applications in computer science, from clustering, ranking and classification, to ridge regression , principal-component analysis and semi-supervised learning (von Luxburg, 2007; Ng et al., 2002; Zhu, 2005a;b; Liu et al., 2019). Given a set of $n$ points in $\mathbb{R}^d$ and a nonnegative function $\mathsf{K} : \mathbb{R}^d \times \mathbb{R}^d \to \mathbb{R}_{\geq 0}$, a kernel matrix has the form that the $i, j$-th entry in the matrix is $\mathsf{K}(x_i, x_j)$.

Kernel matrices and kernel linear-systems naturally arise in modern machine learning and optimization tasks, from Kernel PCA and ridge regression (Alaoui & Mahoney, 2015; Avron et al., 2017a;b; Lee et al., 2020), to Gaussian-process regression (GPR) (Rasmussen & Nickisch, 2010)), federated learning (Konečnỳ et al., 2016), and the 'state-space model' (SSM) in deep learning (Gu et al., 2021b;a). In most of these applications, the underlying data points $x_i$ are dynamically changing across iterations, either by nature or by design, and therefore computational efficiency of numerical linear-algebraic operations in this setting requires dynamic algorithms to maintain the kernel matrix under insertions and deletions of data points.

One motivating application of dynamic linear algebra on geometric graphs is dynamically maintaining a *spectral clustering* (Ng et al., 2002) of the kernel matrix of a weighted graph. In the static setting, a common approach for spectral clustering is *spectral sparsification* (Spielman & Srivastava, 2011; Ng et al., 2002), i.e., to run a spectral clustering algorithm on top of a spectral-sparsifier for the Laplacian matrix of the weighted graph. This approach, however, fails to extend to the dynamic setting, where a small fraction of data points are continually changing, since rebuilding the spectral-sparsifier is prohibitively expensive – Changing a single point $x_i \in P$ changes *an entire row* of $\mathsf{K}(x_i, x_j)$.

Another motivation, arising in statistical physics and astronomy, is the N-body simulation problem (Trenti & Hut, 2008). The problem asks to efficiently simulate a dynamical system of particles, usually under the influence of physical forces, such as gravity. Let $P \subset \mathbb{R}^d$ denote a set of points. In our terminology, this setup corresponds to maintaining, for each $i \in [d]$, a graph $G_i$ on the points in $P$, and letting $C_g$ denote the gravitational constant, and $m_x$ denote the mass of point $x \in P$. Hence, for any two points $u, v \in P$, the non-negative weight/kernel function of the edge $(u, v)$ is defined as $\mathsf{K}_i(u, v) := \left(\frac{C_g \cdot m_u \cdot m_v}{\|u-v\|_2^2}\right) \cdot \left(\frac{|v_i - u_i|}{\|u-v\|_2}\right)$ Denoting the weighted adjacency matrix of $G_i$ by $A_i$, computing the force between the points in the static setting corresponds to $A_i \mathbf{1}$. Once again, this approach (Trenti & Hut, 2008) fails to extend to the dynamic setting in which the $n$-bodies are slowly moving over time, since re-computing $\mathsf{K}()$ would take $\Omega(n)$ time.

Finally, we mention an application to *semi-supervised learning* tasks, where the goal is to extend a partial function, whose values are known only on a subset of the training data, to the entire domain, such that the weighted sum of differences over the set is minimized (Zhu, 2005b)[1]. In the static setting of geometric kernels, this least-squares minimizer can be found by solving a (Laplacian) linear system on $P$. Extending this approach to the dynamic setting requires dynamic spectral sparsifiers.

As mentioned above, one of the main tools for fast linear algebra on geometric graphs is spectral sparsification. Alman, Chu, Schild and Song (Alman et al., 2020) presented a static algorithm for constructing an $\epsilon$-spectral sparsifier on a geometric graph $H = H(G)$ such that $(1 - \epsilon)L_G \preceq L_H \preceq (1 + \epsilon)L_G$ in almost linear time, which avoids explicitly writing the underlying $n \times n$ dense Kernel matrix, and facilitates several basic linear-algebraic operations on geometric graphs, in $\widetilde{O}(n \log^d n) \ll n^2$ time, when the dimension $d$ is fixed. The main goal of this paper is to extend the toolbox of (Alman et al., 2020) to the *dynamic* setting, as motivated by the above applications. More formally:

> *Given a set of $n$ points $P \subset \mathbb{R}^d$ and a function $\mathsf{K}$, is there a dynamic algorithm that can update the spectral sparsifier of the geometric graph $G$ on $P$ and $\mathsf{K}$ in $n^{o(1)}$ time, where in each iteration the location of a point $x_i \in P$ is changed? Is it possible to maintain approximate matrix-vector queries w.r.t the Laplacian matrix of a dynamic geometric graph, and an approximate-inverse of the Laplacian, in $n^{o(1)}$ time?*

Prior to this work, no nontrivial dynamization algorithms were known for geometric graphs for the above linear-algebra primitives. While fully-dynamic $(1 \pm \epsilon)$-spectral edge sparsifiers for general graphs are known (Abraham et al., 2016) (in amortized $\mathrm{poly}(\log(n), 1/\epsilon)$ update time), the setting of geometric graphs is fundamentally different, since as mentioned earlier in the introduction, each update of a point $x_i \in P$ results in an *entire row* update of $\mathsf{K}$, i.e., $O(n)$ edges, making (Abraham et al., 2016) too slow.

The main technical contribution of this paper is a new *dynamic well separated pair decomposition* (WSPD, Definition B.15, (Fischer & Har-Peled, 2005; Alman et al., 2020)). The core of this data structure is a *smooth resampling* technique for efficiently maintaining a WSPD under point-location updates, with mild weight-increase to the sparsifier, by *reusing* randomness (in an adversarially-robust manner).

## 1.1 MODEL

Before we state our main results, let us formally state the dynamic model of geometric graphs we consider:

**Definition 1.1** (Dynamic spectral sparsifier of geometric graph). *Given a set of points $P \subset \mathbb{R}^d$ and kernel function $\mathsf{K} : \mathbb{R}^d \times \mathbb{R}^d \to \mathbb{R}_{\geq 0}$. Let $G$ denote the geometric graph on to $P$ with edge weight $w(x_i, x_j) := \mathsf{K}(x_i, x_j)$. Let $L_G$ be the Laplacian matrix of graph $G$. Let $\epsilon \in (0, 0.1)$ denote an accuracy parameter. We want to design a data structure that dynamically maintains a $(1 \pm \epsilon)$-spectral sparsifier for $G$ and supports the following operations:*

- INITIALIZE$(P \subset \mathbb{R}^d, \epsilon \in (0, 0.1))$, *this operation takes point set $P$ and constructs a $(1 \pm \epsilon)$-spectral sparsifier of $L_G$.*

- UPDATE$(i \in [n], z \in \mathbb{R}^d)$, *this operation takes a vector $z$ as input, and to replace $x_i$ (in point set $P$) by $z$, in the meanwhile, we want this update to be fast and the change in the spectral sparsifier to be small.*

For the above problem, we focus our attention on kernel functions with a natural property called $(C, L)$-multiplicatively Lipschitz. For $C \geq 1$ and $L \geq 1$, we say a function is $(C, L)$-Lipschitz

---

[1]Formally, we are given a function $f : P \to \mathbb{R}$ together with its value on some subset $X \subset P$. Then we aim to extend the function $f$ to the whole set $P$, which can minimize $\sum_{u,v \in P, u \neq v} \mathsf{K}(u, v)(f(u) - f(v))^2$

if that, for all $c \in [1/C, C]$, it holds that $\frac{1}{c^L} \leq \frac{f(cx)}{f(x)} \leq c^L$. Common $(C, L)$-Lipschitz kernels including piecewise exponential kernels, polynomial functions of distance (with non-negative coefficients), and rational function kernels (Alman et al., 2020).

We formally define the sketch task of matrix multiplication here.

**Definition 1.2** (Sketch of approximation to matrix multiplication)**.** *Given a geometric graph $G$ with respect to point set $P$ and kernel function $\mathsf{K}$, and an n-dimensional vector $x$, we want to maintain a low dimensional sketch of an approximation to the multiplication result $L_G x$, where an $\epsilon$-approximation to multiplication result $L_G x$ is a vector $b$ such that $\|b - L_G x\|_2 \leq \epsilon \cdot \|L_G\|_F \cdot \|x\|_2$.*

We also give the formal definition of the sketching approximation to Laplacian solving here.

**Definition 1.3** (Sketch of approximation to Laplacian solving)**.** *Given a geometric graph $G$ with respect to point set $P$ and kernel function $\mathsf{K}$, and an $n$-dimensional vector $b$, we want to maintain a low dimensional sketch of an approximation to the multiplication result $L_G^\dagger b$, i.e., a vector $\widetilde{z}$ such that $\|\widetilde{z} - L_G^\dagger b\|_2 \leq \epsilon \cdot \|L_G^\dagger\|_F \cdot \|b\|_2$*

We here explain the necessity of maintaining a sketch of an approximation instead of the directly maintaining the multiplication result in the dynamic regime. Let the underlying geometry graph on $n$ vertices be $G$ and the vector be $v \in \mathbb{R}^n$. When a $d$-dimensional point is moved in the geometric graph, a column and a row are changed $L_G$. We can assume the first row and first column are changed with no loss of generality. When this happens, if the first entry of $v$ is not 0, all entries will change in the multiplication result. Therefore, it takes at least $\Omega(n)$ time to update the multiplication result exactly. In order to spend subpolynomial time to maintain the multiplication result, we need to reduce the dimension of vectors. Therefore, we use a sketch matrix with $m = \operatorname{poly}\log(n)$ rows.

## 2 OUR RESULTS

Our first main result is a dynamic sparsifier for geometric graphs, with subpolynomial update time in the oblivious adversary model.

**Theorem 2.1** (Informal version of Theorem E.3)**.** *Let $\mathsf{K}$ denote a $(C, L)$-multiplicative Lipschitz kernel function. For any given data point set $P \subset \mathbb{R}^d$ with size $n$, there is a randomized dynamic algorithm DYNAMICGEOSPAR that receives updates of locations of points in $P$ one at a time, and maintains an almost linear spectral sparsifier in $n^{o(1)}$ time with probability $1 - 1/\operatorname{poly}(n)$.*

By introducing additional assumptions regarding dimensions, we can generate outcomes for an adversarial setting.

**Theorem 2.2** (Informal version of Theorem I.5)**.** *Let $\mathsf{K}$ denote a $(C, L)$-multiplicative Lipschitz kernel function. For any given data point set $P \subset \mathbb{R}^d$ with size $n$. Define $\alpha := \frac{\max_{x,y \in P} \|x-y\|_2}{\min_{x,y \in P} \|x-y\|_2}$. If $\alpha^d = O(\operatorname{poly}(n))$, then there is a randomized dynamic algorithm that receives updates of locations of points in $P$ one at a time, and maintains a almost linear spectral sparsifier in $n^{o(1)}$ time with probability $1 - 1/\operatorname{poly}(n)$. It also supports adversarial updates.*

For the dynamic matrix-vector multiplication problem, we give an algorithm to maintain a sketch of the multiplication between the Laplacian matrix of a geometric graph and a given vector in subpolynomial time.

**Theorem 2.3** (Informal version of Theorem F.1)**.** *Let $G$ be a $(C, L)$-Lipschitz geometric graph on $n$ points. Let $v$ be a vector in $\mathbb{R}^n$. There exists an data structure MULTIPLY that maintains a vector $\widetilde{z}$ that is a low dimensional sketch of an approximation to the multiplication result $L_G \cdot v$. MULTIPLY supports the following operations: Part 1. UPDATEG$(x_i, z)$: move a point from $x_i$ to $z$ and thus changing $\mathsf{K}_G$ and update the sketch. This takes $n^{o(1)}$ time. Part 2. UPDATEV$(\delta_v)$: change $v$ to $v + \delta_v$ and update the sketch. This takes $n^{o(1)}$ time. Part 3. QUERY: return the up-to-date sketch.*

We also present a dynamic algorithm to maintain the sketch of the solution to a Laplacian system.

**Theorem 2.4** (Informal version of Theorem G.1)**.** *Let $G$ be a $(C, L)$-Lipschitz geometric graph on $n$ points. Let $b$ be a vector in $\mathbb{R}^n$. There exists an data structure SOLVE that maintains a vector $\widetilde{z}$ that is a low dimensional sketch of an $\epsilon$-approximation to the multiplication result $L_G^\dagger \cdot b$. It supports*

*the following operations: Part 1.* UPDATEG$(x_i, z)$*: move a point from $x_i$ to $z$ and thus changing* K$_G$ *and update the sketch. This takes $n^{o(1)}$ time. Part 2.* UPDATEB$(\delta_b)$*: change $b$ to $b + \delta_b$ and update the sketch. This takes $n^{o(1)}$ time. Part 3.* QUERY*: return the up-to-date sketch.*

**Roadmap.** We provide an overview of techniques used in Section 3. For all the formal proofs, we leave them to the Appendix. We conclude our work in Section 4.

## 3 TECHNICAL OVERVIEW

### 3.1 NOTATIONS

For any two sets $A, B$, we use $A \triangle B$ to denote $(A \backslash B) \cup (B \backslash A)$. Given two symmetric matrices $A, B$, we say $A \preceq B$ if $\forall x, x^\top A x \leq x^\top B x$. For a vector $x$, we use $\|x\|_2$ to denote its $\ell_2$ norm. For psd matrix $A$, we use $A^\dagger$ to denote the pseudo inverse of $A$. For two point sets $A, B$, we denote the complete bipartite graph on $A$ and $B$ by Biclique$(A, B)$. We use $\mu_{n,i}$ to denote an elementary unit vector in $\mathbb{R}^n$ with $i$-th entry 1 and others 0. We use $\mathcal{T}_{\mathrm{mat}}(a, b, c)$ to denote the running time of computing the product of two matrices in the shape of $\mathbb{R}^{a \times b}$ and $\mathbb{R}^{b \times c}$ respectively. For a matrix $A \in \mathbb{R}^{m \times n}$, we use $\|A\|_F$ to denote its Frobenius norm, i.e., $\|A\|_F := (\sum_{i \in [m], j \in [n]} A_{i,j}^2)^{1/2}$. For a matrix, we use $A^\dagger$ to denote the pseudo inverse of matrix $A$. For a vector $x$, and a psd matrix $A$, we use $\|x\|_A := (x^\top A x)^{1/2}$.

### 3.2 FULLY DYNAMIC KERNEL SPARSIFICATION DATA STRUCTURE

A geometric graph w.r.t. kernel function K $: \mathbb{R}^d \times \mathbb{R}^d \to \mathbb{R}$ and points $x_1, \ldots, x_n \in \mathbb{R}^d$ is a graph on $x_1, \ldots, x_n$ where the weight of the edge between $x_i$ and $x_j$ is K$(x_i, x_j)$. An update to a geometric graph occurs when the location of one of these points changes.

In a geometric graph, when an update occurs, the weights of $O(n)$ edges change. Therefore, directly applying the existing algorithms for dynamic spectral sparsifiers ((Abraham et al., 2016)) to update the geometric graph spectral sparsifier will take $\Omega(n)$ time per update. However by using the fact that the points are located in $\mathbb{R}^d$ and exploiting the properties of the kernel function, we can achieve faster update.

Before presenting our dynamic data structure, we first give a high level idea of the static construction of the geometric spectral sparsifier, which is presented in (Alman et al., 2020).

#### 3.2.1 BUILDING BLOCKS OF THE SPARSIFIER

In order to construct a spectral sparsifier more efficiently, one can partition the graph into several subgraphs such that the edge weights on each subgraph are close. On each of these subgraphs, leverage score sampling, which is introduced in (Spielman & Srivastava, 2011) and used for constructing sparsifiers, can be approximated by uniform sampling.

For a geometric graph built from a $d$-dimensional point set $P$, under the assumption that each edge weight is obtained from a $(C, L)$-Lipschitz kernel function (Definition B.1), each edge weight in the geometric graph is not distorted by a lot from the euclidean distance between the two points (Lemma B.22). Therefore, we can compute this partition efficiently by finding a well separated pair decomposition ((Callahan & Kosaraju, 1995), WSPD, Definition B.15) of the given point set.

An $s$-WSPD of $P$ is a collection of pairs $(A_i, B_i)$ of subsets of $P$, such that for all $a \neq b \in P$, there exists a unique $i$ satisfying $a \in A_i, b \in B_i$, and the distance between $A_i$ and $B_i$ (as point sets) is at least $s$ times the diameters of $A_i$ and $B_i$ ($(A_i, B_i)$ is a $s$-well separated (WS) pair). In this case, the distance between point sets $A_i$ and $B_i$ is a $(1 \pm 1/s)$-multiplicative approximation of the distance between any point in $A_i$ and any point in $B_i$.

Each WS pair in the WSPD can be viewed as an unweighted biclique, where the two point sets are the two sides of the bipartite graph. On an unweighted biclique, uniform random sampling and leverage score sampling are equivalent. Therefore, a uniformly random sample of the biclique forms a spectral sparsifier of the biclique, and the union of the sampled edges from all bicliques form a spectral sparsifier of the geometric graph.

However, the time needed for constructing a WSPD depends exponentially on the ambient dimension of the point set and thus WSPD cannot be computed efficiently when the dimension is high. To solve this problem, one can use the ultra low dimensional Johnson Lindenstrauss (JL) projection to project the point set down to $k = o(\log n)$ dimension, such that with high probability, the distance distortion (multiplicative difference between the distance between two points and the distance between their low dimensional images) between any pair of points is at most $n^{C_{jl} \cdot (1/k)}$, where $C_{jl}$ is a universal constant for the JL projection. This distortion becomes an overestimation of the leverage score in the resulting biclique, and can be compensated by sampling $n^{C_{jl} \cdot (1/k)}$ edges.

Then one can perform a 2-WSPD on the $k$-dimensional points. Since JL projection gives a bijection between the $d$-dimensional points and their $k$-dimensional images, a 2-WSPD of the $k$-dimensional point set gives us a $(2 \cdot n^{C_{jl}/k})$-WSPD of the $d$-dimensional point set $P$. The $d$-dimensional bicliques resulted from this $(2 \cdot n^{C_{jl}/k})$-WSPD of $P$ is what we use to sample edges and construct the sparsifier.

In summary, after receiving a set of points $P$, we use the ultra-low dimensional JL projection to project these points to a $k = o(\log n)$ dimensional space, run a 2-WSPD on the $k$-dimensional points, and then map the $k$-dimensional WSPD result back to the $d$-dimensional point set to obtain a $(2 \cdot n^{C_{jl}(1/k)})$-WSPD of $P$. For each pair $(A, B)$ in this $d$-dimensional WSPD, we randomly sample edges from Biclique$(A, B)$. The union of all sampled edges is a spectral sparsifier of the geometric graph on $P$.

### 3.2.2 DYNAMIC UPDATE OF THE GEOMETRIC SPECTRAL SPARSIFIER.

For a geometric graph build on a point set $P$, we want the above spectral sparsifier to be able to handle the following update[2] :

Point location change $(x_i \in P, z \in \mathbb{R}^d)$: move the point from location $x_i$ to location $z$. This is equivalent to removing point $x \in P$ and then adding $z$ to $P$.

However, in order to update the geometric spectral sparsifier efficiently, there are a few barriers that we need to overcome.

**Updating WSPD** When the point set $P$ changes, we want to update the 2-WSPD such that the number of WS pairs that are changed in the WSPD is small. (Fischer & Har-Peled, 2005) presented an algorithm to update the list of WS pairs, but it cannot be used directly in this situation, because the (Fischer & Har-Peled, 2005) algorithm is only able to return a list of WS pairs such that the singleton containing the inserted (or removed) point is one of the vertex subsets in these pairs. However, in order to use the up-to-date WSPD to update the sparsifier, we need to know not only the WS pairs $(A, B)$ where $A$ or $B$ is a singleton consisting of the inserted (or removed) point, but also all other WS pairs $(A, B)$ such that the inserted (or removed) point is in $A$ or $B$. The (Fischer & Har-Peled, 2005) algorithm is not able to do this.

Fortunately, one $s$-WSPD construction algorithm presented in (Har-Peled, 2011) has the property that each point $x$ appears only in $s^{O(d)} O(\log \alpha)$ WS pairs. This allows us to find all WS pairs affected by a point location change in $2^{O(k)} O(\log \alpha)$ time, since we are maintaining a 2-WSPD and the dimension of the point set is $k$. We summarize this algorithm below. The detailed discussion of the WSPD update can be found in Section E.4.

The (Har-Peled, 2011) WSPD algorithm constructs a *compressed quadtree* associated with the point set $P$, and the WS pairs are pairs of nodes of the compressed quadtree. To summarize, a *quadtree* is a hierarchical partition of a $k$-dimensional hypercube enclosing $P$. It is obtained by recursively dividing the $k$-dimensional region into $2^k$ smaller regions called *cells* (divide equally along each axis), which can be further subdivided until each resulting cell contains only one point. The tree representing this hierarchy is called a quadtree and each cell in this hierarchy corresponds to a tree node of the quadtree. The cells containing only one point are the *leaf nodes* of the tree. In a quadtree, there can be a long chain of tree nodes that contain the same set of points. We replace this chain by the first and last nodes on the chain, and an edge between them. The resulted tree is the compressed

---

[2] We assume that throughout the update, the aspect ratio of the point set, denoted by $\alpha = \frac{\max_{x,y \in P} \|x-y\|_2}{\min_{x,y \in P} \|x-y\|_2}$, does not change.

quadtree associated with $P$. The compressed quadtree has size $O(|P|)$, and supports the following operations in $O(\log n)$ time: (1) finding the leaf node that contains a given point $x$, or the parent node under which the leaf node containing $x$ should be inserted if $x \notin P$, (2) inserting a leaf node containing a given point $x$, and (3) removing a leaf node containing a given point $x$.

The WSPD is a list of pairs of well separated compressed quadtree nodes. For efficient update, we let the WSPD data structure to be a container (of WS pairs) that supports looking up all WS pairs containing a tree node $n$ for a given $n$ in time linear in the size of output.

When a point location update occurs, suppose point $x_i$ is moved to $z$. We can do the following to find all WS pairs that need to be updated.

- Use the compressed quad tree data structure to locate leaf nodes that contains $x_i$ and $z$ (since $z$ is not in the point set before the update, we locate the parent node under which $z$ should be inserted)
- Go from each of these leaf nodes to the root of the compressed quad tree, for each tree node $n$ visited in this process, use the WSPD data structure to find all WS pairs containing $n$.
- Update all WS pairs found in the previous step and the compressed quadtree.

Algorithm 6 in Section E.4 is a detailed version of this WSPD update scheme.

**Resampling from bicliques.**    After updating the WSPD, we want to generate a uniform sample of edges from the new biclique. We show that with high probability, this can be done in $n^{o(1)}$ time with high probability.

When a point location change happens and point $x_i$ is moved to $z$, each pair $(A, B)$ in the WSPD list will undergo one and only one of the following changes, Part 1. Remaining $(A, B)$. Part 2. Becoming $(A \backslash \{x_i\}, B)$ or $(A, B \backslash \{x_i\})$. Part 3. Becoming $(A \cup \{z\}, B)$ or $(A, B \cup \{z\})$. Part 4. Becoming $(A \backslash \{x_i\} \cup \{z\}, B)$, $(A, B \backslash \{x_i\} \cup \{z\})$, $(A \backslash \{x_i\}, B \cup \{z\})$ or $(A \cup \{z\}, B \backslash \{x_i\})$.

For each WS pair $(A, B)$ that remains $(A, B)$, we do not need to do anything about it. For each WS pair $(A, B)$ that is changed $(A', B')$, in order to maintain a spectral sparsifier of Biclique$(A', B')$, we need to find a new uniform sample from Biclique$(A', B')$. Simply drawing another uniform sample from $A' \times B'$ cannot be done fast enough when $|A' \times B'|$ is large and this resampling will cause a lot of edge weight changes in the final sparsifier, which is not optimal.

To overcome this barrier, suppose after an update, a WS pair $(A, B)$ is changed to $(A', B')$. Since the size difference between $A$ and $A'$ and the size difference between $B$ and $B'$ are at most constant, the size of $(A' \times B') \cap (A \times B)$ is much larger than the size of $(A' \times B') \backslash (A \times B)$. Therefore, when we draw a uniform sample from $A' \times B'$, most of the edges in the sample should be drawn from $(A' \times B') \cap (A \times B)$. Since we already have a uniform sample $E$ from $A \times B$, which contains a uniform sample from $(A' \times B') \cap (A \times B)$, we can reuse $E$ in the following way:

Let $H = E \cap (A' \times B')$. For each edge that needs to be samples, we flip an unfair coin for which the probability of landing on head is $\frac{|(A' \times B') \cap (A \times B)|}{|A' \times B'|}$, and we do the following (See Figure 2 for a visual example):

- If the coin lands on head, we sample an edge from $H$ without repetition;
- Otherwise we sample an edge from $(A' \times B') \backslash (A \times B)$ without repetition.

Algorithm 7 in Section E.5 is a detailed version of this resampling scheme. With properly set probability for the coin flip, doing the sampling this way generates a uniform sample of $A' \times B'$, and with high probability, the difference between the new sample and $E$ is small.

However, in this process, although the difference between the new sample and $E$ is small, we still need to flip a coin for each new sample point. When the sample size is big, this can be slow.

The running time of resampling can be improved by removing a small number of edges from $E$. Indeed, suppose we want to resample $s$ edges from $A' \times B'$, the number of edges that need to be drawn from $(A' \times B') \backslash (A \times B)$ follows a Binomial distribution with parameters $s$ and $\frac{|(A' \times B') \backslash (A \times B)|}{|A' \times B'|}$. We have the following improved resampling algorithm:

Let $H = E \cap (A' \times B')$.

- Generate a random number $x$ under $\text{Binomial}(s, \frac{|(A' \times B')\backslash(A \times B)|}{|A' \times B'|})$.

- Remove $x + |H| - s$ pairs from $H$.

- Sample $x$ new edges uniformly from $(A' \times B')\backslash(A \times B)$ and add them to $H$.

Since $x$ has $n^{o(1)}$ expected value, with high probability (Markov inequality), $x$ is $n^{o(1)}$, the difference between $E$ and the new sample is $n^{o(1)}$, and the resampling process can be done in $n^{o(1)}$ time. We omitted the edge case where the size of $H$ is less than $s - x$. Algorithm 8 in Section E.6 is a detailed version of this sublinear resampling scheme.

**Dynamic update.** Combining the above, we can update the spectral sparsifier (see Section E.7 for details). When a point location update occurs, suppose point $x_i$ is moved to $z$. We use the ultra low dimensional JL projection matrix to find the $O(k)$-dimensional images of $x_i$ and $z$. Then we update the $O(k)$-dimensional WSPD. For each $O(k)$-dimensional modified pair in the WSPD, we find the corresponding $d$-dimensional modified pairs, and resample edges from these $d$-dimensional modified pairs to update the spectral sparsifier. Since there are $2^{O(k)} \log \alpha$ modified pairs in each update and for each modified pair, with probability $1 - \delta$, the uniform sample can be update in $O(\delta^{-1}\epsilon^{-2}n^{o(1)})$ time, the dynamic update can be completed in $O(\delta^{-1}\epsilon^{-2}n^{o(1)} \log \alpha)$ time per update.

### 3.3 Adaptive Adversarial Updates

The dynamic algorithm above is only able to handle oblivious updates. Recall the building blocks of the dynamic update algorithms. We compute the JL projection of the update points, update the WSPD for the low dimensional projections, and resample from the corresponding $d$-dimensional bicliques. Among these steps, the WSPD update algorithm is deterministic; the resampling algorithm uses fresh randomness for every round of updates. Therefore, the only building part that can be exploited by an adaptive adversary is the JL projection. Below in this overview, we explain how we achieve a JL distance estimation against adaptive adversaries. In this section, we provide an overview of techniques we use for adversarial analysis.

#### 3.3.1 Adversarial Distance Estimation

Let a random vector $V = (V_1, V_2, \cdots, V_d) \in \mathbb{R}^d$ be sampled from Gaussian distribution and $U = \frac{1}{\|V\|}V$ be the normalized vector. Let vector $Z = (U_1, U_2, \cdots, U_k) \in \mathbb{R}^k$ be the projection of $U$ onto the first $k$ components. From the properties of random variables sampled from Gaussian distribution, we can compute $\Pr[d(U_1^2 + \cdots + U_k^2) \leq k\beta(U_1^2 + \cdots + U_d^2)]$ via algebraic manipulations. Let $L = \|Z\|^2$. We show that when $\beta < 1$, we have $\Pr[L \leq \frac{\beta k}{d}] \leq \exp(\frac{k}{2}(1 - \beta + \ln \beta))$ and when $\beta \geq 1$, we have $\Pr[L \geq \frac{\beta k}{d}] \leq \exp(\frac{k}{2}(1 - \beta + \ln \beta))$. By carefully choosing $\beta = n^{-2c/k} < 1$, we can prove $\Pr[L \leq \frac{\beta k}{d}] \leq n^{-c}$. And when $\beta = n^{1/k}$, we can prove $\Pr[L \geq \frac{\beta k}{d}] \leq \exp(-\log^{1.9} n)$.

With the above analysis in hand, we can prove that there exists a map $f : \mathbb{R}^d \to \mathbb{R}^k$ such that for each fixed points $u, v \in \mathbb{R}^d$, we have $\|u - v\|_2^2 \leq \|f(u) - f(v)\|_2^2 \leq \exp(c_0 \cdot \sqrt{\log n})\|u - v\|_2^2$ with high success probability. We design a $\epsilon_0$-net of $\{x \in \mathbb{R}^d \mid \|x\|_2 \leq 1\}$ denoted as $N$ which contains $|N| \leq (10/\epsilon_0)^{O(\log n)}$ points (Here we assume $d = O(\log n)$). Then we prove that for all net points, the approximation guarantee still holds with high success probability via union bound. Finally, we want to generalize the distance estimation approximation guarantee to all points on the unit ball by quantizing the off-net point to its nearest on-net point. After rescaling the constant, we can obtain the same approximation guarantee with high probability.

Given a set of data points $\{x_i\}_{i=1}^n$, and a sketching matrix $\Pi \in \mathbb{R}^{k \times d}$ defined in Definition H.1, we initialize a set of precomputed projected data points $\widetilde{x}_i = \Pi \cdot x_i$. To answer the approximate distance between a query point and all points in the data structure, we compute the distance as $u_i = n^{1/k} \cdot \sqrt{d/k} \cdot \|\widetilde{x}_i - \Pi q\|_2$ and prove it provides $\exp(\Theta(\sqrt{\log n}))$-approximation guarantee against adversarially chosen queries. When we need to update the $i$-th data point with a new vector $z \in \mathbb{R}^d$, we update $\widetilde{x}_i$ with $\Pi \cdot x_i$.

### 3.3.2 SPARSIFIER WITH ROBUSTNESS TO ADVERSARIAL UPDATES

With the estimation robust for adversarial query, we are able to get a spectral sparsifier which supports adversarial updates of points, by applying the data structure in the construction of sparsifier (Setting the sketching dimension to be $O(\sqrt{\log n})$). Here we provide overview of our design to make it possible.

**Net argument.** In order to make the distance estimation robust, one needs to argue that, for arbitrary point, it has high probability to have high precision. The data structure we use for distance estimation has a failure probability of $n^{-c}$, where $c$ is a constant we can set to be small. We can build an $\epsilon$-net $N$ with size of $|N| = \mathrm{poly}(n)$. Then by union bound over the net, the failure probability of distance estimation on the net is bounded by $n^{O(1)-c}$. Then by triangle inequality, we directly get the succeed probability guarantee for arbitrary point queries.

**$\alpha$ and $d$ induce the size of the net.** From the discussion above, we note that, in order to make the $\epsilon$-net sufficient for union bound, it must have the size of $\mathrm{poly}(n)$. From another direction, we need to make that, all the points in the set are distinguishable in the nets, i.e., for two different points $A, B \in \mathbb{R}^d$, the closest points of the net to $A$ and $B$ are different. To make sure this, we must set the gap $\epsilon_1$ of the net to be less than the minimum distance of the points in the set. Without loss of generality, we first make the assumption that, all the points are in the $\ell_2$ unit ball of $\mathbb{R}^d$, i.e., the set $\{x \in \mathbb{R}^d \mid \|x\|_2 \leq 1\}$. Then by the definition of aspect ratio $\alpha := \frac{\max_{x,y \in P} d(x,y)}{\min_{x',y' \in P} d(x',y')}$, the minimum distance of the points in $P$ is $1/\alpha$. Thus, when we set the gap $\epsilon_1 \leq C \cdot \alpha^{-1}$ for some constant $C$ small enough, every pair of points $x, y \in P$ is distinguishable in the net. Then there are $O(\alpha^d)$ points in the net of the $\ell_2$ unit ball in $\mathbb{R}^d$ (See Figure 3).

**Balancing the aspect ratio and dimension.** By the above paragraph, we know the set size is $O(\alpha^d)$ to make the points distinguishable. Recall that, our distance estimation data structure has failure probability of $n^{-c}$. And in order to make the union bound sufficient for our net, we need to apply it over the $|N|^2$ pairs from $N$. That is, to make the total failure probability sufficient, we need to restrict $|N| = \mathrm{poly}(n)$. And in the former paragraphs, we already know that $|N| = O(\alpha^d)$, thus we have the balancing constraint of the aspect ratio and dimension $\alpha^d = O(\mathrm{poly}(n))$.

### 3.4 MAINTAINING A SKETCH OF AN APPROXIMATION TO LAPLACIAN MATRIX MULTIPLICATION

Let $M$ be an $n \times n$ matrix and $x$ be a vector in $\mathbb{R}^n$. We say a vector $b$ is an $\epsilon$-approximation to $Mx$ if $\|b - Mx\|_{M^\dagger} \leq \epsilon \|Mx\|_{M^\dagger}$. Note that $\|x\|_A := \sqrt{x^\top Ax}$. Let $G$ be a graph and $H$ be a $\epsilon$-spectral sparsifier of $G$. By definition, this means $(1-\epsilon)L_G \preceq L_H \preceq (1+\epsilon)L_G$. Note that, if $A$ is a symmetric PSD matrix and symmetric $B$ is a matrix such that $(1-\epsilon)A \preceq B \preceq (1+\epsilon)A$, then we have $\|Bv - Av\|_{A^\dagger} \leq \epsilon \|Av\|_{A^\dagger}$ holds for all $v$. Then, we have: Let $G$ be a graph on $n$ vertices and $H$ be a $\epsilon$-spectral sparsifier of $G$. For any $v \in \mathbb{R}^n$, $L_H v$ is an $\epsilon$-approximation of $L_G v$. Thus, to maintain a sketch of an $\epsilon$-approximation of $L_G x$, it suffices to maintain a sketch of $L_H x$.

The high level idea is to combine the spectral sparsifier defined in Section E and a sketch matrix to compute a sketch of the multiplication result $L_H v$ and try to maintain this sketch when the graph and the vector change.

We here justify the decision of maintaining a sketch instead of the directly maintaining the multiplication result. Let the underlying geometry graph on $n$ vertices be $G$ and the vector be $v \in \mathbb{R}^n$. When a point is moved in the geometric graph, a column and a row are changed in $L_G$. We can assume the first row and first column are changed with no loss of generality. When this happens, if the first entry of $v$ is not 0, all entries will change in the multiplication result. Therefore, it takes at least $\Omega(n)$ time to update the multiplication result. In order to spend subpolynomial time to maintain the multiplication result, we need to reduce the dimension of vectors. Therefore, we use a sketch matrix (with $m = \epsilon^{-2} \log(n/\delta)$ rows, see Lemma F.3 for details) to project vectors down to lower dimensions.

**Maintaining the multiplication result efficiently.** In order to speed up the update, we generate two independent sketches $\Phi$ and $\Psi$, and maintain a sketch of $L_H$, denoted by $\widetilde{L}_H = \Phi L_H \Psi^\top$ and a sketch of $v$ denoted by $\widetilde{v} = \Psi v$. Since $\Phi$ and $\Psi$ are generated independently, in expectation $\Phi L_H \Psi^\top \Psi v = \Phi L_H v$. We store this result as the sketch.

Our spectral sparsifier has the property that with high probability, each update to the geometric graph $G$ incurs only a sparse changes in the sparsifier $H$, and this update can be computed efficiently. Therefore, when an update occurs to $G$, $\Delta L_H$ is sparse, so $\Phi \Delta L_H \Psi^\top$ can be computed efficiently. We use $\Phi \Delta L_H \Psi^\top$ to update the sketch. When a sparse update occurs to $v$, $\Psi \Delta v$ can be computed efficiently. Since $\widetilde{L}_H$ and $\Psi \Delta v$ are $m$-dimensional operator and vector, $\widetilde{L}_H \Psi \Delta v$ can be computed efficiently. We use $\widetilde{L}_H \Psi \Delta v$ to update the sketch.

### 3.5 Maintaining a sketch of an approximation to the solution of a Laplacian system

We start with another folklore fact:

**Fact 3.1** (folklore). *If* $(1-\epsilon)L_G \preceq L_H \preceq (1+\epsilon)L_G$, *then we have* $(1-2\epsilon)L_G^\dagger \preceq L_H^\dagger \preceq (1+2\epsilon)L_G^\dagger$.

Let $G$ be a graph on $n$ vertices and $H$ be a $\epsilon$-spectral sparsifier of $G$. For any vector $b$, $L_H^\dagger b$ is an $\epsilon$-approximation of $L_G^\dagger b$. Thus, to maintain a sketch of an $\epsilon$-approximation of $L_G^\dagger x$, it suffices to maintain a sketch of $L_H^\dagger x$. The high level idea is again to combine the spectral sparsifier defined in Section E and a sketch matrix to compute a sketch of the multiplication result $L_H^\dagger v$ and try to maintain this sketch when the graph and the vector change.

**Caveat: using a different sketch.** When trying to maintain a sketch of a solution to $L_H x = b$, [3] the canonical way of doing this is to maintain $\overline{x}$ such that $\Phi L_H \overline{x} = \Phi b$. However, here $\overline{x}$ is still an $n$-dimensional vector and we want to maintain a sketch with lower dimension. Therefore, we apply another sketch $\Psi$ to $\overline{x}$ and maintain $\widetilde{x}$ such that $\Phi L_H \Psi^\top \widetilde{x} = \Phi b$.

**Maintaining the inversion result efficiently.** We maintain a sketch of $L_H$, denoted by $\widetilde{L}_H = \Phi L_H \Psi^\top$ and a sketch of $b$ denoted by $\widetilde{b} = \Phi b$. Since $\widetilde{L}_H$ is a $m$-dimensional operator, its pseudoinverse can be computed efficiently in $m^\omega$ time, where $\omega$ is the matrix multiplication constant. We use $\widetilde{L}_H^\dagger$ to denote the pseudoinverse of $\widetilde{L}_H$, and compute $\widetilde{L}_H^\dagger \cdot \widetilde{b}$. We store this multiplication result as the sketch. Our spectral sparsifier has the property that with high probability, each update to the geometric graph $G$ incurs only a sparse changes in the sparsifier $H$, and this update can be computed efficiently. Therefore, when an update occurs to $G$, $\Delta L_H$ is sparse, so $\Phi \Delta L_H \Psi^\top$ can be computed efficiently. We use $\Phi \Delta L_H \Psi$ to update the $\widetilde{L}_H$ and recompute $\widetilde{L}_H^\dagger$. We then update the sketch to $L_H^\dagger \cdot \widetilde{b}$ with the updated $L_H^\dagger$. When a sparse update occurs to $b$, $\Phi \Delta b$ can be computed efficiently. Since $\widetilde{L}_H^\dagger$ and $\Phi \Delta b$ are $m$-dimensional operator and vector, $\widetilde{L}_H^\dagger \Phi \Delta b$ can be computed efficiently. We use $\widetilde{L}_H^\dagger \Psi \Delta b$ to update the sketch.

## 4 Conclusion

In this work, we present dynamic algorithms for maintaining geometric graphs efficiently. Our main contributions include the introduction of the DYNAMICGEOSPAR data structure and techniques for handling adversarial queries and low-dimensional sketches with near-optimal initialization and update times, significantly improving existing methods. By combining spectral sparsification and Johnson-Lindenstrauss projections, we ensur efficient recomputation of graph structures with sparse changes. We prove that our data structure can dynamically maintain a $(1 \pm \epsilon)$-spectral sparsifier with high probability, leverage JL projections to maintain low-dimensional sketches for efficient updates and queries, and design algorithms that are robust against adaptive adversarial queries. Our work has significant practical implications for real-time updates in geometric graphs.

---

[3] Although $L_H$ is sparse, its pseudoinverse $L_H^\dagger$ has no guarantee to be sparse (see bottom of page 113 in (Saad, 2003)).

## ETHIC STATEMENT

This paper does not involve human subjects, personally identifiable data, or sensitive applications. We do not foresee direct ethical risks. We follow the ICLR Code of Ethics and affirm that all aspects of this research comply with the principles of fairness, transparency, and integrity.

## REPRODUCIBILITY STATEMENT

We ensure reproducibility of our theoretical results by including all formal assumptions, definitions, and complete proofs in the appendix. The main text states each theorem clearly and refers to the detailed proofs. No external data or software is required.

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

# Appendix

**Roadmap.** We divide the appendix as follows. Section A provides related work of our paper. Section B gives the preliminary for our paper. Section C discusses the sketching techniques we use. Section D provide the full main algorithm. Section E gives the fully dynamic spectral sparsifier for geometric graphs. Section F gives our sketch data structure for matrix multiplication. Section G introduces the algorithm for solving Laplacian system. Section H introduces the distance estimation data structure supporting adversarial queries. Based on that, Section I gives our spectral sparsifier that is robust to adaptive adversary. Section J provides some figure to better explain this work. Section **??** discusses the limitation of this work. Section **??** provides a elaborate discussion about potential societal impact.

## A    RELATED WORK

**Dynamic Sparsifier.**    There has been some work focused on maintaining the dynamic sparsifier in a efficient time (Durfee et al., 2019). Their follow-up work (Gao et al., 2022; Brand et al., 2022) provides an algorithm for computing exact maximum flows on graphs with bounded integer edge capacities. Quanrud's work on spectral sparsification of graphs with metrics and kernels (Quanrud, 2021) provide efficient algorithm for constructing an sparsifier for the graphs.

**Solvers of Laplacian System.**    For a Laplacian linear system of a graph with $m$ edges, it is a widely-studied problem (Spielman & Teng, 2004; Koutis et al., 2014; 2011; Kelner et al., 2013; Lee & Sidford, 2013; Cohen et al., 2014; Kyng et al., 2016; Kyng & Sachdeva, 2016). It has been shown that it can be solved in time $\widetilde{O}(m \log(1/\epsilon))$. While the existing algorithm is very fast when the graph is sparse enough, we should focused on faster algorithms since our target graph might be dense.

**Approximating the Kernel Density Function (KDF).**    There are some works (Charikar & Siminelakis, 2017; Backurs et al., 2018) studying algorithms for kernel density function. (Charikar & Siminelakis, 2017) studied the Kernel Density Estimate (KDE) problem and they gave an efficient data structure such that, given a data set with a specific kernel function, it approximates the kernel density of a query point in sublinear time. Later work (Backurs et al., 2018) presented a collection of algorithms for KDF approximating the "smooth" kernel functions. (Zandieh et al., 2023; Alman & Song, 2023) shows how to use kernel technique to compute attention matrix in large language models.

**Kernel Functions.**    Kernel method is a popular technique in data analysis and machine learning (Souza, 2010). The most popular and widely-used kernel functions are in the form of $\mathsf{K}(x, y) = f(\|x - y\|_2)$. We list some of them here: the Gaussian kernel (Ng et al., 2002; Rahimi & Recht, 2007), multiquadric kernel (Beatson & Greengard, 1997), circular kernel (Boughorbel et al., 2005), power kernel (Fleuret et al., 2003), log kernel (Beatson & Greengard, 1997; Martinsson, 2012) and inverse multiquadric kernel (Micchelli, 1984; Martinsson, 2012).

**Dynamic Algorithms used in Optimization.**    In addition, dynamic algorithms have been widely used in many of the optimization tasks. Usually, most optimization analysis are robust against noises and errors, such as linear programming (Cohen et al., 2019; Jiang et al., 2021), empirical risk minimization (Lee et al., 2019; Qin et al., 2023), semi-definite programming (Huang et al., 2021; Gu & Song, 2022), general programming (Deng et al., 2023), integral optimization (Jiang et al., 2023), training neural network (Brand et al., 2021; Song et al., 2021a;b), and sum of squares method (Jiang et al., 2022). Approximate solutions are sufficient to for these optimizations.

## B    PRELIMINARY

### B.1    DEFINITIONS

We define the $(C, L)$-Lipschitz function as follows:

**Definition B.1.** *For $C \geq 1$ and $L \geq 1$, a function is $(C, L)$-Lipschitz if for all $c \in [1/C, C]$,*

$$\frac{1}{c^L} \leq \frac{f(cx)}{f(x)} \leq c^L.$$

We define the *Laplacian* of a graph:

**Definition B.2** (Laplacian of graph). *Let $G = (V, E, w)$ be a connected weighted undirected graph with $n$ vertices and $m$ edges, together with a positive weight function $w : E \to \mathbb{R}_+$. If we orient the edges of $G$ arbitrarily, we can write its Laplacian as*

$$L_G = A^\top W A,$$

*where $A \in \mathbb{R}^{m \times n}$ is the signed edge-vertex incidence matrix, given by*

$$A(e, v) = \begin{cases} 1, & \text{if } v \text{ is the head of } e \\ -1 & \text{if } v \text{ is the tail of } e \\ 0 & \text{otherwise} \end{cases}$$

*and $W \in \mathbb{R}_+^{m \times m}$ is the diagonal matrix such that $W(e, e) = w(e)$, for all $e \in E$. We use $\{a_e\}_{e \in E}$ to denote the row vectors of $A$.*

It follows obviously that $L_G$ is positive semidefinite since for any $x \in \mathbb{R}^n$,

$$x^\top L_G x = x^\top A^\top W A x = \|W^{1/2} A x\|_2^2 \geq 0.$$

Since $L_G$ is symmetric, we can diagnolize it and write

$$L_G = \sum_{i=1}^{n-1} \lambda_i u_i u_i^\top,$$

where $\lambda_1, \ldots, \lambda_{n-1}$ are the nonzero eigenvalues of $L_G$ and $u_1, \ldots, u_{n-1}$ are the corresponding orthonormal eigenvectors. The *Moore-Penrose Pseudoinverse* of $L_G$ is

$$L_G^\dagger := \sum_{i=1}^{n-1} \frac{1}{\lambda_i} u_i u_i^\top.$$

### B.2 BASIC ALGEBRA

**Fact B.3** (Folklore). *Let $\epsilon \in (0, 1/2)$. Given two positive semidefinite matrix $A \in \mathbb{R}^{n \times n}$ and $B \in \mathbb{R}^{n \times n}$ such that*

$$(1 - \epsilon)A \preceq B \preceq (1 + \epsilon)A,$$

*then we have:*

- *Part 1. $(1 + \epsilon)^{-1} A^\dagger \preceq B^\dagger \preceq (1 - \epsilon)^{-1} A^\dagger$.*

- *Part 2. $\|Bx - Ax\|_{A^\dagger} \leq \epsilon \|Ax\|_{A^\dagger}, \forall x \in \mathbb{R}^n$.*

*Proof.* **Proof of Part 1.** The first statement follows from Fact B.4 directly.

**Proof of Part 2.**

$$\|Bx - Ax\|_{A^\dagger}^2 = x^\top (B - A) A^\dagger (B - A) x$$

$$\epsilon^2 \cdot \|Ax\|_{A^\dagger}^2 = \epsilon^2 \cdot x^\top A A^\dagger A x$$

It is obvious that

$$-\epsilon A \preceq B - A \preceq \epsilon A$$

Thus by Fact B.5, we have

$$(B - A)A^{\dagger}(B - A) \preceq \epsilon^2 A A^{\dagger} A.$$

Thus we have for any $x \in \mathbb{R}^n$,

$$\begin{aligned}
\|Bx - Ax\|_{A^{\dagger}}^2 &= x^{\top}(B - A)A^{\dagger}(B - A)x \\
&\leq \epsilon^2 \cdot x^{\top}(B - A)A^{\dagger}(B - A)x \\
&= \epsilon^2 \cdot \|Ax\|_{A^{\dagger}}^2.
\end{aligned}$$

Thus we complete the proof. $\qquad\square$

**Fact B.4.** *If $A \preceq B$, then $B^{\dagger} \preceq A^{\dagger}$.*

*Proof.* We denote the SVD of $A$ and $B$ by $A = U_A \Sigma_A V_A^{\top}$ and $B = U_B \Sigma_B V_B^{\top}$, then we have for any $x \in \mathbb{R}^n$,

$$\begin{aligned}
x^{\top}(B^{\dagger} - A^{\dagger})x &= x^{\top}(V_B^{\top}\Sigma_B^{-1}U_B - V_A^{\top}\Sigma_A^{-1}U_A)x \\
&= x^{\top}V_B^{\top}\Sigma_B^{-1}U_Bx - x^{\top}V_A^{\top}\Sigma_A^{-1}U_Ax \\
&= (x^{\top}U_B^{\top}\Sigma_B V_B x)^{-1} - (x^{\top}U_A^{\top}\Sigma_A V_A x)^{-1} \\
&= (x^{\top}Bx)^{-1} - (x^{\top}Ax)^{-1} \\
&\geq 0,
\end{aligned}$$

where the last step follows from $A \preceq B$. Thus we complete the proof. $\qquad\square$

**Fact B.5.** *Let $A, C$ denote two psd matrices. Let $B$ be a symmetric matrix. Suppose $-C \preceq B \preceq C$, then we have*

$$BAB \preceq CAC$$

### B.3 JOHNSON-LINDERSTRAUSS TRANSFORM

#### B.3.1 ULTRA-LOW DIMENSION JL

**Lemma B.6** (Ultra-low Dimensional Projection (Johnson & Lindenstrauss, 1984; Dasgupta & Gupta, 2003))**.** *For $k = o(\log n)$, with high probability at least $1 - 1/\operatorname{poly}(n)$ the maximum distortion in pairwise distance obtained from projecting $n$ points into $k$ dimensions (with appropriate scaling) is at most $n^{O(1/k)}$, e.g.,*

$$\|x - y\|_2 \leq \|f(x) - f(y)\|_2 \leq n^{O(1/k)} \cdot \|x - y\|_2$$

*where $f$ is the projection from $\mathbb{R}^d$ to $\mathbb{R}^k$.*

Throughout this paper, we use $C_{\mathrm{jl}}$ to denote the constant on the exponent, i.e. the distortion is bounded above by $n^{C_{\mathrm{jl}} \cdot (1/k)}$.

**Lemma B.7** ((Johnson & Lindenstrauss, 1984; Dasgupta & Gupta, 2003))**.** *Let $k < d$. Let $V_1, V_2, \cdots, V_d$ be $d$ independent Gaussian $N(0, 1)$ random variables, $V = (V_1, V_2, \cdots, V_d)$, and let $U = \frac{1}{\|V\|}V$. Let the vector $Z = (U_1, U_2, \cdots, U_k) \in \mathbb{R}^k$ be the projection of $U$ onto the first $k$ components and let $L = \|Z\|^2$ be the square of the norm of $Z$. Then*

- *Part 1. If $\beta < 1$, then*

$$\Pr[L \leq \frac{\beta k}{d}] \leq \beta^{k/2} \cdot (1 + \frac{(1-\beta)k}{(d-k)})^{(d-k)/2} \leq \exp(\frac{k}{2}(1 - \beta + \ln\beta))$$

- *Part 2. If $\beta > 1$, then*

$$\Pr[L \geq \frac{\beta k}{d}] \leq \beta^{k/2} \cdot (1 + \frac{(1-\beta)k}{(d-k)})^{(d-k)/2} \leq \exp(\frac{k}{2}(1 - \beta + \ln\beta))$$

### B.3.2 USEFUL LEMMAS ON JL

Using Lemma B.7, we can show that

**Lemma B.8.** *Let $\beta = n^{1/k}$, then we have $\Pr[L \geq \frac{\beta k}{d}] \leq \exp(-\frac{1}{4}kn^{1/k}) \leq \exp(-\log^{1.9} n)$.*

*Proof.* We show that

$$\Pr[L \geq \frac{\beta k}{d}] \leq \exp(\frac{k}{2}(1 - \beta + \ln \beta))$$

$$\leq \exp(-\frac{k}{2}\frac{\beta}{2})$$

$$= \exp(-\frac{k}{2}\frac{n^{1/k}}{2})$$

$$\leq \exp(-\log^{1.9} n)$$

where the first step follows from Lemma B.7, the second step follows from $\beta/2 \geq 1 + \ln \beta$, the third step follows from $\beta = n^{1/k}$, and the last step follows from $n^{1/k} \geq \log^{1.9} n$. $\square$

Using Lemma B.7 and choosing parameter $\beta$ carefully, we can show that:

**Lemma B.9.** *Let $\beta = n^{-2c/k}/e = 2^{-2c(\log n)/k}/e < 1$, then we have $\Pr[L \leq \frac{\beta k}{d}] \leq n^{-c}$.*

*Proof.* We show that

$$\Pr[L \leq \frac{\beta k}{d}] \leq \beta^{k/2} \cdot (1 + \frac{(1-\beta)k}{(d-k)})^{(d-k)/2}$$

$$= \beta^{k/2} \cdot (1 + \frac{(1-\beta)k}{(d-k)})^{\frac{(d-k)}{k(1-\beta)} \cdot \frac{k(1-\beta)}{2}}$$

$$\leq \beta^{k/2} \cdot e^{k/2}$$

$$\leq (\beta e)^{k/2}$$

$$\leq n^{-c}$$

where the first step comes from Lemma B.7, the second step follows that $(d - k)/2 = \frac{(d-k)}{k(1-\beta)} \cdot \frac{k(1-\beta)}{2}$, the third step follows $(1 + \frac{1}{a})^a = e$ and $\frac{k(1-\beta)}{2} \leq \frac{k}{2}$, the fourth step simplifies the term, and the last step follows from $\beta = n^{-2c/k}/e$.

$\square$

## B.4 WELL SEPARATED PAIR DECOMPOSITION (WSPD)

We assume that throughout the process, all points land in $[0, 1]^d$ and the *aspect ratio* of the point set is at most $\alpha$.

**Definition B.10.** *The aspect ratio ($\alpha$) of a point set $P$ is*

$$\alpha := \frac{\max_{x,y \in P} d(x, y)}{\min_{x',y' \in P} d(x', y')}.$$

We state several standard definitions from literature (Callahan & Kosaraju, 1995).

**Definition B.11** (Bounding rectangle). *Let $P \subset \mathbb{R}^d$ be a set of points, we define the bounding rectangle of $P$, denoted as $R(P)$, to be the smallest rectangle in $\mathbb{R}^d$ such that encloses all points in $P$, where "rectangle" means some cartesian product $[x_1, x'_1] \times [x_2, x'_2] \times \cdots \times [x_d, x'_d] \in \mathbb{R}^n$. For all $i \in [d]$, We define the length of $R$ in $i$-th dimension by $l_i(R) := x'_i - x_i$. We denote $l_{\max}(R) := \max_{i \in [d]} l_i(R)$ and $l_{\min}(R) := \min_{i \in [d]} l_i(R)$. When $l_i(R)$ are all equal for $i \in [d]$, we say $R$ is a d-cube, and denote its length by $l(R)$. For any set of points $P \subseteq \mathbb{R}^d$, we denote $l_i(P) = l_i(R(P))$.*

**Definition B.12** (Well separated point sets). *Point sets $P, Q$ are well separated with separation $s$ if $R(P)$ and $R(Q)$ can be contained in two balls of radius $r$, and the distance between these two balls is at least $s \cdot r$, where we say $s$ is the* separation.

**Definition B.13** (Interaction product). *The interaction product of point sets $P, Q$, denoted by $P \otimes Q$ is defined as*

$$P \otimes Q := \{\{p, q\} \mid p \in P, q \in Q, p \neq q\}$$

**Definition B.14** (Well separated realization). *Let $P, Q \subseteq \mathbb{R}^d$ be two sets of points. A well separated realization of $P \otimes Q$ is a set $\{(P_1, Q_1), \dots, (P_k, Q_k)\}$ such that*

1. *$P_i \subset P$, $Q_i \subset Q$ for all $i \in [k]$.*

2. *$P_i \cap Q_i = \emptyset$ for all $i \in [k]$.*

3. *$P \otimes Q = \bigcup_{i=1}^{k} P_i \otimes Q_i$.*

4. *$P_i$ and $Q_i$ are well-separated.*

5. *$(P_i \otimes Q_i) \cap (P_j \otimes Q_j) = \emptyset$ for $i \neq j$.*

Throughout the paper, we will mention that a set $P$ is *associated with* a binary tree $T$. Here we mean the tree $T$ has leaves labeled by a set containing only one point which is in $P$. All the non-leaf nodes are labeled by the union of the sets labeled with its subtree.

Given set $P \subseteq \mathbb{R}^d$, let $T$ be a binary tree associated with $P$. For $A, B \subseteq P$, we say that a realization of $A \otimes B$ *uses* $T$ if all the $A_i$ and $B_i$ in the realization are nodes in $T$.

**Definition B.15** (Well separated pair decomposition). *A well separated pair decomposition (WSPD) of a point set $P$ is a structure consisting of a binary tree $T$ associated with $P$ and a well separated realization of $P \otimes P$ uses $T$.*

The result of (Callahan & Kosaraju, 1995; Har-Peled, 2011) states that for a point set $P$ of $n$ points, a well separated pair decomposition of $P$ of $O(n)$ pairs can be computed in $O(n \log n)$ time. There are two steps of computing a well separated decomposition: (1) build compressed quad tree (defined below in Definition B.17) for the given point set; (2) find well separated pairs from the tree (Algorithm 1).

**Definition B.16** (Quad tree). *Given a point set $P \subset [0, 1)^d$, a tree structure $\mathcal{T}$ can be constructed in the following way:*

- *The root of $\mathcal{T}$ is the region $[0, 1)^d$*

- *For each tree node $n \in \mathcal{T}$, we can obtain $2^d$ subregions by equally dividing $n$ into two halves along each of the $d$ axes. The children of $n$ in $\mathcal{T}$ are the subregions that contain points in $P$. $n$ has at most $2^d$ children.*

- *The dividing stops when there is only one point in the cell.*

In a quad tree, we define the **degree** of a tree node to be the number of children it has. There can be a lot of nodes in $T$ that has degree 1. Particularly, there can be a path of degree one nodes. Every node on this path contain the same point set. To reduce the size of the quad tree, we compress these degree one paths.

**Definition B.17** (Compressed quad tree). *Given a quad tree $T$, for each a path of degree one nodes, we replace it with the first and last nodes on the path, with one edge between them. We call this resulting tree a compressed quad tree.*

**Lemma B.18** (Chapter 2 in (Har-Peled, 2011)). *The compressed quad tree data structure $T$ has the following properties:*

- *Given a point $p$, $p$ exists in at most $O(\log \alpha)$ quad tree nodes.*

- *The height of the tree is $O(\min(n, \log \alpha))$.*

*$T$ supports the following operations:*

- QTFASTPL$(T, p)$ *returns the leaf node containing* $p$, *or the parent node under which* $\{p\}$ *should be inserted if* $p$ *does not exist in* $T$, *in* $O(\log n)$ *time.*

- QTINSERTP$(T, p)$ *adds* $p$ *to the* $T$ *in* $O(\log n)$ *time.*

- QTDELETEP$(T, p)$ *removes* $p$ *from* $T$ *in* $O(\log n)$ *time.*

**Lemma B.19** (Theorem 2.2.3 in (Har-Peled, 2011)). *Given a* $d$-*dimensional point set* $P$ *of size* $n$, *a compressed quad tree of* $P$ *can be constructed in* $O(dn \log n)$ *time.*

---

**Algorithm 1** Finding well separated pairs

---

1: **procedure** WSPD$(Q, u, v)$         ▷ $Q$ is a compressed quad tree, $u, v$ are tree nodes on $Q$, Lemma B.21
2:     **if** $u$ and $v$ are well separated **then**
3:         **return** $(u, v)$
4:     **else**
5:         **if** $l_{\max}(u) > l_{\max}(v)$ **then**
6:             Let $u_1, \ldots, u_m$ denote the children of $u$
7:             **return** $\bigcup_i$ WSPD$(Q, u_i, v)$
8:         **else**
9:             Let $v_1, \ldots, v_m$ denote the children of $v$
10:            **return** $\bigcup_i$ WSPD$(Q, u, v_i)$
11:         **end if**
12:     **end if**
13: **end procedure**
14:
15: **procedure** COMPUTEWSPD$(Q)$         ▷ $Q$ is a compressed quad tree
16:     $r \leftarrow$ root of $Q$
17:     **return** WSPD$(Q, r, r)$
18: **end procedure**

---

**Theorem B.20** ((Callahan & Kosaraju, 1995)). *For point set* $P \subseteq \mathbb{R}^d$ *of size* $n$ *and* $s > 1$, *a* $s$-*WSPD of size* $O(s^d n)$ *can be found in* $O(s^d n + n \log n)$ *time and each point is in at most* $2^{O(d)} \log \alpha$ *pairs.*

**Lemma B.21** ((Callahan & Kosaraju, 1995; Har-Peled, 2011)). *Given a compressed quad tree* $Q$ *of* $n$ *points in* $\mathbb{R}^d$, *two nodes* $u, v$ *in the tree* $Q$. *The procedure* COMPUTEWSPD *(Algorithm 1) generates the WSPD from the tree* $Q$, *and runs in time*

$$O(2^d \cdot n \log n).$$

### B.5 PROPERTIES OF $(C, L)$-LIPSCHITZ FUNCTIONS

**Lemma B.22** (Lemma 6.8 in (Alman et al., 2020)). *Let* $G$ *be a graph and* $G'$ *be another graph on the same set of vertices with different edge weights satisfying*

$$\frac{1}{K} w_{G'}(e) \leq w_G(e) \leq K w_{G'}(e)$$

*Let* $f : \mathbb{R} \to \mathbb{R}$ *be a* $(C, L)$-*lipschitz kernel function (Definition B.1), for some* $C < K$. *Let* $f(G)$ *be the graph obtained by switching each edge weight from* $w(e)$ *to* $f(w(e))$. *Then,*

$$\frac{1}{K^{2L}} w_{G'}(e) \leq w_G(e) \leq K^{2L} w_{G'}(e).$$

### B.6 LEVERAGE SCORE AND EFFECTIVE RESISTANCE

**Definition B.23.** *Given a matrix* $A \in \mathbb{R}^{m \times n}$, *we define* $\sigma \in \mathbb{R}^m$ *to denote the leverage score of* $A$, *i.e.,*

$$\sigma_i = a_i^\top (A^\top A) a_i, \forall i \in [m].$$

Let $(G, V, E)$ be an graph obtained by arbitrarily orienting the edges of an undirected graph, with $n$ points and $m$ edges, together with a weight function $w : E \to \mathbb{R}_+$. We now describe the electrical flows on the graph. We let vector $I_{\text{ext}} \in \mathbb{R}^n$ denote the currents injected at the vertices. Let $I_{\text{edge}} \in \mathbb{R}^m$ denote currents induced in the edges (in the direction of orientation) and $V \in \mathbb{R}^n$ denotes the potentials induced at the vertices. Let $A, W$ be defined as Definition B.2. By Kirchoff's current law, the sum of the currents entering a vertex is equal to the amount injected at the vertex, i.e.,

$$A^\top I_{\text{edge}} = I_{\text{ext}}.$$

By Ohm's law, the current flow in an edge is equal to the potential difference across its ends times its conductance, i.e.,

$$I_{\text{edge}} = WAV.$$

Combining the above, we have that

$$I_{\text{ext}} = A^\top (WAV) = L_G V.$$

If $I_{\text{ext}} \perp \text{Span}(1_n) = \ker(K)$, that is, the total amount of current injected is equal to the total amount extracted, then we have that

$$V = L_G^\dagger I_{\text{ext}}.$$

**Definition B.24** (Leverage score of a edge in a graph). *We define the* effective resistance *or* leverage score *between two vertices* u *and* v *to be the potential difference between them when a unit current is injected at one that extracted at the other.*

**Lemma B.25** (Algebraic form of leverage score, (Spielman & Srivastava, 2011)). *Let* $(G, E, V)$ *be a graph described as above, for any edge* $e \in E$, *the leverage score (effective resistance) of* $e$ *has the following form*

$$R(e) = AL_G^\dagger A^\top (e, e),$$

*where the matrix* $A, L_G$ *is defined as Definition B.2.*

*Proof.* We now derive an algebraic expression for the effective resistance in terms of $L_G^\dagger$. For a edge $e \in E$, we use $R(e)$ to denote its effective resistance. To inject and extract a unit current across the endpoints of an edge $(u, v)$, we set $I_{\text{ext}} = a_e^\top$, which is clearly orthogonal to $1_n$. The potentials induced by $I_{\text{ext}}$ at the vertices are given by $V = L_G^\dagger a_e^\top$. To measure the potential difference across $e = (u, v)$, we simply multiply by $a_e$ on the left:

$$V_v - V_u = (\mu_{n,v} - \mu_{n,u})^\top V = a_e L_G^\dagger a_e^\top.$$

It follows that, the effective resistance across $e$ is given by $a_e L_G^\dagger a_e^\top$ and that the matrix $AL_G^\dagger A^\top$ has its diagonal entries $AL_G^\dagger A^\top (e, e) = R(e)$. $\qquad\square$

### B.7 SPECTRAL SPARSIFIER

Here we give the formal definition of spectral sparsifier of a graph:

**Definition B.26** (Spectral sparsifier). *Given an arbitrary undirected graph* $G$, *let* $L_G$ *denote the Laplacian (Definition B.2) of* $G$. *We say* $H$ *is a* $\epsilon$-spectral sparsifier of $G$ if

$$(1 - \epsilon)L_G \preceq L_H \preceq (1 + \epsilon)L_G.$$

## C SKETCHING TECHNIQUES

### C.1 DEFINITIONS

We first introduce the formal definition of the sparse embedding matrix:

**Definition C.1** (Sparse Embedding matrix (Nelson & Nguyên, 2013)). *Let* $h : [n] \times [s] \to [b/s]$ *be a random 2-wise independent hash function and* $\sigma : [n] \times [s] \to \{+1, -1\}$ *be a random 4-wise independent hash function. Then* $R \in \mathbb{R}^{b \times n}$ *is a sparse embedding matrix with parameter* $s$ *if we set* $R_{(j-1)b/s+h(i,j),i} = \sigma(i,j)/s$ *for all* $(i, j) \in [n] \times [s]$ *and all other entries to zero.*

We also define the JL-moment property:

**Definition C.2** (JL-moment property, Definition 12 in (Woodruff, 2014)). *We say a distribution $\mathcal{D}$ on matrices $S \in \mathbb{R}^{k \times d}$ has the $(\epsilon, \delta, \ell)$-JL moment property if that for all $x \in \mathbb{R}^d$ with $\|x\|_2 = 1$, it holds that*

$$\mathop{\mathbb{E}}_{S \sim \mathcal{D}}[|\|Sx\|_2^2 - 1|^\ell] \leq \epsilon^\ell \cdot \delta.$$

We give the formal definition of approximating matrix product:

**Definition C.3** (Approximating matrix product (AMP) (Kane & Nelson, 2012; Woodruff, 2014)). *Let $\epsilon \in (0, 1)$ be a precision parameter. Let $\delta \in (0, 1)$ be the failure probability. Given any two matrix $A, B$ each with $n$ rows, we say a randomized matrix $R \in \mathbb{R}^{b \times n}$ from a distribution $\Pi$ satisfies $(\epsilon, \delta)$-approximate matrix product of $A$ and $B$ if*

$$\mathop{\Pr}_{R \sim \Pi}[\|A^\top R^\top R B - A^\top B\|_F > \epsilon \cdot \|A\|_F \cdot \|B\|_F] \leq \delta.$$

## C.2 Useful Results of Sparse Embedding Matrix

Here in this section, we introduce the following technical theorem from literature, which gives the concentration property of the sparse embedding matrices.

**Lemma C.4** (Theorem 19 in (Kane & Nelson, 2012)[4]). *Let $\epsilon, \delta \in (0, 1)$ be two parameters. Let $\mathcal{D}$ be a distribution over $d$ columns that satisfies the $(\epsilon, \delta, \ell)$-JL moment property for some $\ell \geq 2$. Then for two matrices $A, B$ with $n$ rows, it holds that*

$$\mathop{\Pr}_{\Psi \sim \mathcal{D}}[\|A^\top \Psi^\top \Psi B - A^\top B\|_F > 3 \cdot \epsilon \cdot \|A\|_F \cdot \|B\|_F] \leq \delta.$$

There is a result giving the JL-moment property of sparse embedding matrices in literature.

**Lemma C.5** (Implicitly[5] in (Cohen et al., 2015)). *The sparse embedding matrix (Definition C.1) with $m = O(\epsilon^{-2} \cdot \log(1/\delta))$ and $s = \Omega(\epsilon^{-1} \cdot \log(1/\delta))$ satisfies $(\epsilon, \delta, \log(1/\delta))$-JL moment property.*

Now we give the AMP property of the sparse embedding matrix.

**Lemma C.6** (AMP of Sparse Embedding matrix). *Let $A \in \mathbb{R}^{n \times d_A}$ and $B \in \mathbb{R}^{n \times d_B}$ be two arbitrary matrices. Let $R \in \mathbb{R}^{m \times n}$ be a Sparse Embedding matrix as defined in Definition C.1 with $m = O(\epsilon^{-2} \cdot \log(1/\delta))$ and $s = \Omega(\epsilon^{-1} \cdot \log(1/\delta))$ non-zero entries of each column, then it satisfies $(\epsilon, \delta)$-AMP of $A$ and $B$, and $A^\top R^\top R B$ can be computed in time*

$$s \cdot \mathrm{nnz}(A) + s \cdot \mathrm{nnz}(B) + \mathcal{T}_{\mathrm{mat}}(d_A, m, d_B).$$

*Proof.* By Lemma C.5, $R$ satisfy $(\epsilon, \delta, \log(1/\delta))$-JL moment property. Since $\log(1/\delta) > 2$ trivially holds, then by Lemma C.4, we proved the correctness of the lemma. It takes $s \cdot \mathrm{nnz}(A)$ time to compute $A^\top R^\top$, $s \cdot \mathrm{nnz}(B)$ time to compute $RB$, and $\mathcal{T}_{\mathrm{mat}}(d_A, m, d_B)$ to compute $A^\top R^\top R B$. □

# D Algorithm

Here in this section, we give our main algorithm as follows. The main theorem of the algorithm and the analysis with respect to the correctness and running time can be found in Appendix F.

# E Fully Dynamic Spectral Sparsifier for Geometric Graphs in Sublinear Time

A geometric graph w.r.t. kernel function $\mathsf{K} : \mathbb{R}^d \times \mathbb{R}^d \to \mathbb{R}$ and points $x_1, \ldots, x_n \in \mathbb{R}^d$ is a graph on $x_1, \ldots, x_n$ where the weight of the edge between $x_i$ and $x_j$ is $\mathsf{K}(x_i, x_j)$. An update to a geometric graph occurs when the location of one of these points changes.

---

[4]For examples, see Theorem 17 in (Nelson & Nguyên, 2013) and Theorem 13 in (Woodruff, 2014)]

[5]See Remark 2 at page 9 of (Cohen et al., 2015)

---

**Algorithm 2** Maintaining a sketch of an approximation to the solution to a Laplacian equation

---

1: **data structure** SOLVE          ▷ Theorem 2.4 and Theorem G.1
2: **members**
3:     DYNAMICGEOSPAR dgs          ▷ This is the sparsifier $H$
4:     $\Phi, \Psi \in \mathbb{R}^{m \times n}$: two independent sketching matrices
5:     $\widetilde{L} \in \mathbb{R}^{m \times m}$          ▷ A sketch of $L_H$
6:     $\widetilde{L}^\dagger \in \mathbb{R}^{m \times m}$          ▷ A sketch of $L_H^\dagger$
7:     $\widetilde{b} \in \mathbb{R}^m$          ▷ A sketch of $b$
8:     $\widetilde{z} \in \mathbb{R}^m$          ▷ A sketch of the multiplication result
9: **EndMembers**
10:
11: **procedure** INIT($x_1, \cdots, x_n \in \mathbb{R}^d, b \in \mathbb{R}^n$)
12:     Initialize $\Phi, \Psi$
13:     dgs.INITIALIZE($x_1, \ldots, x_n$)
14:     $\widetilde{b} \leftarrow \Phi b$
15:     $\widetilde{L} \leftarrow \Phi \cdot$ dgs.GETLAPLACIAN() $\cdot \Psi^\top$
16:     $\widetilde{L}^\dagger \leftarrow$ PSEUDOINVERSE($\widetilde{L}$)
17:     $\widetilde{z} \leftarrow \widetilde{L}^\dagger \cdot \widetilde{b}$
18: **end procedure**
19:
20: **procedure** UPDATEG($x_i, z \in \mathbb{R}^d$)
21:     dgs.UPDATE($x_i, z$)
22:     $\widetilde{L} \leftarrow \widetilde{L} + \Phi \cdot$ dgs.GETDIFF() $\cdot \Psi^\top$
23:     $\widetilde{L}^\dagger_{\text{new}} \leftarrow$ PSEUDOINVERSE($\widetilde{L}$)
24:     $\widetilde{z} \leftarrow \widetilde{L}^\dagger \cdot \widetilde{b}$
25:     $\widetilde{L}^\dagger \leftarrow \widetilde{L}^\dagger_{\text{new}}$
26: **end procedure**
27:
28: **procedure** UPDATEB($\Delta b \in \mathbb{R}^n$)          ▷ $\Delta b$ is sparse
29:     $\Delta \widetilde{b} \leftarrow \Phi \cdot \Delta b$
30:     $\widetilde{z} \leftarrow \widetilde{z} + \widetilde{L}^\dagger \cdot \Delta \widetilde{b}$
31: **end procedure**
32:
33: **procedure** QUERY
34:     **return** $\widetilde{z}$
35: **end procedure**

---

In a geometric graph, when an update occurs, the weights of $O(n)$ edges change. Therefore, directly applying the existing algorithms for dynamic spectral sparsifiers ((Abraham et al., 2016)) to update the geometric graph spectral sparsifier will take $\Omega(n)$ time per update. However by using the fact that the points are located in $\mathbb{R}^d$ and exploiting the properties of the kernel function, we can achieve faster update.

Before presenting our dynamic data structure, we first have a high level idea of the static construction of the geometric spectral sparsifier, which is presented in (Alman et al., 2020).

**Building Blocks of the Sparsifier.** In order to construct a spectral sparsifier more efficiently, one can partition the graph into several subgraphs such that the edge weights on each subgraph are close. On each of these subgraphs, leverage score sampling, which is introduced in (Spielman & Srivastava, 2011) and used for constructing sparsifiers, can be approximated by uniform sampling.

For a geometric graph built from a $d$-dimensional point set $P$, under the assumption that each edge weight is obtained from a $(C, L)$-Lipschitz kernel function (Definition B.1), each edge weight in the geometric graph is not distorted by a lot from the euclidean distance between the two points (Lemma B.22). Therefore, we can compute this partition efficiently by finding a well separated pair decomposition (WSPD, Definition B.15) of the given point set.

A $s$-WSPD of $P$ is a collection of well separated (WS) pairs $(A_i, B_i)$ such that for all $a \neq b \in P$, there is a pair $(A, B)$ satisfying $a \in A, b \in B$, and the distance between $A$ and $B$ is at least $s$ times the diameters of $A$ and $B$ ($A$ and $B$ are $s$-well separated). Therefore, the distance between $A$ and $B$ is a $(1 + 1/s)$-multiplicative approximation of the distance between any point in $A$ and any point in $B$ and each WS pair in the WSPD can be viewed as a unweighted biclique (complete bipartite graph). On an unweighted biclique, uniform random sampling and leverage score sampling are equivalent. Therefore, a uniformly random sample of the biclique forms a spectral sparsifier of the biclique, and union of the sampled edges from all bicliques form a spectral sparsifier of the geometric graph.

However, the time needed for constructing a WSPD is exponentially dependent on the ambient dimension of the point set and thus WSPD cannot be computed efficiently when the dimension is high. To solve this problem, one can use the ultra low dimensional Johnson Lindenstrauss (JL) projection to project the point set down to $k = o(\log n)$ dimension such that with high probability the distance distortion (multiplicative difference between the distance between two points and the distance between their low dimensional images) between any pair of points is at most $n^{C_{\mathrm{jl}} \cdot (1/k)}$, where $C_{\mathrm{jl}}$ is a constant. This distortion becomes an overestimation of the leverage score in the resulting biclique, and can be compensated by sampling $n^{C_{\mathrm{jl}} \cdot (1/k)}$ edges. Then one can perform a 2-WSPD on the $k$-dimensional points. Since JL projection gives a bijection between the $d$-dimensional points and their $k$-dimensional images, a 2-WSPD of the $k$-dimensional point set gives us a canonical $(2 \cdot n^{C_{\mathrm{jl}}(1/k)})$-WSPD of the $d$-dimensional point set $P$. This $(2 \cdot n^{C_{\mathrm{jl}}(1/k)})$-WSPD of $P$ is what we use to construct the sparsifier.

**Dynamic Update of the Geometric Spectral Sparsifier.** We present the following way to update the above sparsifier. In order to do this, we need to update the ultra low dimensional JL projection, the WSPD and the sampled edges from each biclique. In order to update JL projection for $O(n)$ updates, we initialize the JL projection matrix with $O(n)$ points so that with high probability, the distortion is small for $O(n)$ updates.

To update the WSPD, we note that each point appears only in $O(\log \alpha)$ WS pairs and we can find all these pairs in $2^{O(k)} \log \alpha$ time (Section E.4).

To update the sampled edges, Algorithm 8 updates the old sample to a new one such that with high probability, the number of edges changed in the sample is at most $n^{o(1)}$ and this can be done in $n^{o(1)}$ time (Section E.6).

Combining the above, we can update the spectral sparsifier (Section E.7).

Below is the layout of this section. In Section E.1, we provide some definitions. In Section E.2, we define the members of our data structure. In Section E.3, we present the algorithm for initialization. In Section E.4 we state an algorithm to find modified pairs in WSPD when a point's location is changed. In Section E.5, we first propose a (slow) resampling algorithm takes $O(n)$ time to resample $n^{o(1)}$ edges. In Section E.6, we then explain how to improve the running time of (slow) resampling algorithm. In Section E.7, we prove the correctness of our update procedure. In Section E.8, we apply a black box reduction to our update algorithm to obtain a fully dynamic update algorithm.

### E.1 DEFINITIONS

We define our problem as follows:

**Definition E.1** (Restatement of Definition 1.1). *Given a set of points $P \subset \mathbb{R}^d$ and kernel function $\mathsf{K} : \mathbb{R}^d \times \mathbb{R}^d \to \mathbb{R}_{\geq 0}$. Let $G$ denote the geometric graph that is corresponding to $P$ with the $(i, j)$ edge weight is $w_{i,j} := \mathsf{K}(x_i, x_j)$. Let $L_{G,P}$ denote the Laplacian matrix of graph $G$. Let $\epsilon \in (0, 0.1)$ denote an accuracy parameter. The goal is to design a data structure that dynamically maintain a $(1 \pm \epsilon)$-spectral sparsifier for $G$ and supports the following operations:*

- INITIALIZE($P \subset \mathbb{R}^d, \epsilon \in (0, 0.1)$), *this operation takes point set $P$ and constructs a $(1 \pm \epsilon)$-spectral sparsifier of $L_G$.*

- UPDATE($i \in [n], z \in \mathbb{R}^d$), *this operation takes a vector $z$ as input, and to replace $x_i$ (in point set $P$) by $z$, in the meanwhile, we want to spend a small amount of time and a small number of changes to spectral sparsifer so that*

**Definition E.2** (Restatement of Definition B.10). *Given a set of points $P = \{x_1, \cdots, x_n\} \subset \mathbb{R}^d$. We define the aspect ratio $\alpha$ of $P$ to be*

$$\alpha := \frac{\max_{i,j} \|x_i - x_j\|_2}{\min_{i,j} \|x_i - x_j\|_2}.$$

The main result we want to prove in this section is

**Theorem E.3** (Formal version of Theorem 2.1). *Let $\alpha$ be the aspect ratio of a $d$-dimensional point set $P$ defined above. Let $k = o(\log n)$. There exists a data structure DYNAMICGEOSPAR that maintains a $\epsilon$-spectral sparsifier of size $O(n^{1+o(1)})$ for a $(C,L)$-Lipschitz geometric graph such that*

- DYNAMICGEOSPAR *can be initialized in*

$$O(ndk + \epsilon^{-2} n^{1+o(L/k)} \log n \log \alpha)$$

  *time.*

- DYNAMICGEOSPAR *can handle point location changes. For each change in point location, the spectral sparsifier can be updated in*

$$O(dk + 2^{O(k)} \epsilon^{-2} n^{o(1)} \log \alpha)$$

  *time. With high probability, the number of edges changed in the sparsifier is at most*

$$\epsilon^{-2} 2^{O(k)} n^{o(1)} \log \alpha.$$

### E.2 THE GEOMETRIC GRAPH SPECTRAL SPARSIFICATION DATA STRUCTURE

In the following definition, we formally define the members we maintain in the data structure.

**Definition E.4.** *In DYNAMICGEOSPAR, we maintain the following objects:*

- *$P$: a set of points in $\mathbb{R}^d$*

- *$\mathcal{H}$: an $n^{1+o(1)}$ size $\epsilon$-spectral sparsifier of the geometric graph generated by kernel $\mathsf{K}$ and points $P$*

- *$\Pi$: a JL projection matrix*

- *$Q$: the image of $P$ after applying projection $\Pi$*

- *$T$: a quad tree of point set $Q$*

- *$\mathcal{P}$: a WSPD for point set $Q$ obtained from $P$*

- EDGES*: a set of tuples $(A_i, B_i, E_i)$. $E_i$ is a set of edges uniformly sampled from Biclique$(X_i, Y_i)$, where $X_i$ and $Y_i$ are the $d$-dimensional point sets corresponding to $A_i$ and $B_i$ respectively*

### E.3 INITIALIZATION

Here in this section, we assume that kernel function $\mathsf{K}(x,y) = f(\|x - y\|_2^2)$ is $(C,L)$-Lipschitz (Definition B.1).

**Lemma E.5.** *Let $\alpha \geq 0$ be defined as Definition B.10. INITIALIZE$(P \subset \mathbb{R}^d, \epsilon \in (0, 0.1), \delta \in (0, 0.1), \mathsf{K})$ (Algorithm 4) takes a $d$-dimensional point set $P$ as inputs and runs in*

$$O(ndk + \epsilon^{-2} n^{1+O(L/k)} 2^{O(k)} \log n \log \alpha)$$

*time, where $k$ is the JL dimension, $k = o(\log n)$.*

*Proof.* The running time consists of the following parts:

---

**Algorithm 3** Data Structure

---

1: **data structure** DYNAMICGEOSPAR           ▷ Theorem E.3
2: **members**                   ▷ Definition E.4
3:  $\mathcal{H}$             ▷ An $n^{1+o(1)}$ size sparsifier
4:  $P \subset \mathbb{R}^d$          ▷ A point set for the geometric graph
5:  $\Pi \in \mathbb{R}^{k \times d}$            ▷ Projection matrix
6:  $Q \subset \mathbb{R}^k$ ▷ A set of $k = o(\log n)$ dimensional points obtained by applying $\Pi$ to all points in $P$
7:  $T$             ▷ A quad tree generated from $P'$
8:  $\mathcal{P} = \{(A_i, B_i)\}_{i=1}^m$        ▷ A WSPD of $P$ based on $T$
9:  EDGES $= \{(A_i, B_i, E_i)\}_{i=1}^m$  ▷ $E_i$ is a set of edges sampled from biclique $(X_i, Y_i)$, where $X_i$ and $Y_i$ are the $d$-dimensional point sets
10: **end members**
11: **end data structure**

---

**Algorithm 4** DYNAMICGEOSPAR

---

1: **data structure** DYNAMICGEOSPAR           ▷ Theorem E.3
2: **procedure** INITIALIZE($P, \epsilon, \delta, \mathsf{K}$)          ▷ Lemma E.5
3:  $P \leftarrow P$
4:  $\Pi \leftarrow$ a random $(k \times d)$ JL-matrix        ▷ Lemma B.6
5:  $Q \leftarrow \{\Pi \cdot p \mid p \in P\}$
6:  $T \leftarrow$ build a compressed quad tree for $Q$      ▷ Lemma B.19
7:  $\mathcal{P} \leftarrow$ WSPD($T, \text{root}(T), \text{root}(T)$)       ▷ Algorithm 1
8:  EDGES, $\mathcal{H} \leftarrow$ INITSPARSIFIER($\mathcal{P}, \mathsf{K}, \epsilon, k$)    ▷ Algorithm 5
9: **end procedure**
10: **end data structure**

---

- Line 4 and Line 5 takes time $O(ndk)$ to Generate the projection matrix and compute the projected sketch;

- By Lemma B.19, Line 6 takes time

$$O(nk \log n);$$

to build the quad tree.

- By Lemma B.21, Line 7 takes time

$$O(n \times 2^k \log n)$$

to generate the WSPD.

- By Lemma E.7, Line 8 takes time

$$O(\epsilon^{-2} n^{1+O(L/k)} 2^{o(k)} \log n \log \alpha)$$

to generate the sparsifier.

Adding them together we have the total running time is

$$O(ndk + \epsilon^{-2} n^{1+O(L/k)} 2^{o(k)} \log n \log \alpha).$$

Thus we complete the proof.

$\square$

We here state a trivial fact of sampling edges from a graph.

**Fact E.6** (Random sample from a graph). *For any graph $G$ and a positive integer $s \in \mathbb{Z}_+$, there exists a random algorithm* RANDSAMPLE($G, s$) *such that, it takes $G$ and $s$ as inputs, and outputs a set containing $s$ edges which are uniformly sampled from $G$ without replacement. This algorithm runs in time $O(s)$.*

---

**Algorithm 5** DYNAMICGEOSPAR: init sparsifier.

---

1: **data structure**                                             ▷ Theorem E.3 DYNAMICGEOSPAR
2: **procedure** INITSPARSIFER($\mathcal{P}, \mathsf{K}, \epsilon, k$)                                ▷ Lemma E.7
3:     $\mathcal{H} \leftarrow$ empty graph with $n$ vertices
4:     EDGES $\leftarrow \emptyset$
5:     **for** $(A, B) \in \mathcal{P}$ **do**
6:         Find $(X \subseteq P, Y \subseteq P)$ such that $X, Y$ are the $d$-dimensional point sets corresponding
    to $A, B$ respectively.
7:         $G \leftarrow$ BICLIQUE$(\mathsf{K}, X, Y)$
8:         $s \leftarrow \epsilon^{-2} n^{O(L/k)}(|A| + |B|) \log(|A| + |B|)$
9:         $E \leftarrow$ RANDSAMPLE$(G, s)$                                        ▷ Fact E.6
10:         EDGES $\leftarrow$ EDGES $\cup \{(A, B, E)\}$
11:         Normalize edges in $E$ by scale $|A||B|/s$
12:         $\mathcal{H} \leftarrow \mathcal{H} \cup E$
13:     **end for**
14:     **return** EDGES, $\mathcal{H}$
15: **end procedure**
16: **end data structure**

---

Now we are able to introduce the initialization algorithm for the sparsifier.

**Lemma E.7.** *The procedure* INITSPARSIFIER *(Algorithm 5) takes* $\mathcal{P}, \mathsf{K}, \epsilon, k$ *as input, where* $\mathcal{P}$ *is a WSPD of the JL projection of point set* $P$, $\mathsf{K}$ *is a* $(C, L)$-*Lipschitz kernel function,* $k = o(\log n)$ *and* $\epsilon$ *is an error parameter, runs in time*

$$O(\epsilon^{-2} n^{1+O(L/k)} 2^{O(k)} \log n \log \alpha)$$

*and outputs* EDGES, $\mathcal{H}$, *such that*

- EDGES *is the set of tuples such that for each* $(A_i, B_i, E_i) \in$ EDGES, $E_i$ *is a set of edges sampled from Biclique$(A_i, B_i)$.*

- $\mathcal{H}$ *is a* $(1 \pm \epsilon)$- *spectral sparsifier of the* $\mathsf{K}$-*graph based on* $P$

- *the size of* $\mathcal{H}$ *is size* $O(\epsilon^{-2} n^{1+O(L/k)})$

*Proof.* We divide the proof into the following paragraphs.

**Correctness**    We view each well separated pair as a biclique. Since $\mathcal{P}$ is a 2-WSPD on a JL projection of $P$ of distortion at most $n^{O(1/k)}$, by Lemma B.6, for any WS pair $(A, B)$ and its corresponding $d$-dimensional pair $(X, Y)$, we have that

$$\frac{\max_{x \in X, y \in Y} \|x - y\|_2}{\min_{x \in X, y \in Y} \|x - y\|_2} \leq 2 \cdot n^{O(1/k)}.$$

By Lemma B.22, it holds that

$$\frac{\max_{x \in X, y \in Y} \mathsf{K}(\|x - y\|_2)}{\min_{x \in X, y \in Y} \mathsf{K}(\|x - y\|_2)} \leq 2 \cdot n^{O(L/k)}.$$

By seeing the biclique as an unweighted graph where all edge weights are equal to the smallest edge weight, one can achieve a overestimation of the leverage score of each edge. For each edge, the leverage score (Definition B.24) is overestimated by at most

$$O(n^{O(L/k)}(|X| + |Y|)/(|X||Y|)).$$

Therefore, by uniformly sampling

$$s = O(\epsilon^{-2} n^{O(L/k)} \cdot (|X| + |Y|) \cdot \log(|X| + |Y|))$$

edges from $\text{Biclique}(X, Y)$ and normalize the edge weights by $|X||Y|/s$, we obtained a $\epsilon$-spectral sparsifier of $\text{Biclique}(X, Y)$.

Since $\mathcal{H}$ is the union of the sampled edges over all bicliques, $\mathcal{H}$ is a $\epsilon$-spectral sparsifier of $\mathsf{K}_G$ of at most $\epsilon^{-2} n^{1+O(L/k)}$ edges. EDGES stores the sampled edges from each biclique by definition.

**Running time** Since each vertex appears in at most $2^{O(k)} \log \alpha$ different WS pairs (Theorem B.20), the total time needed for sampling is at most

$$\epsilon^{-2} 2^{O(k)} \cdot n^{1+O(L/k)} \log n \log \alpha.$$

Thus we complete the proof. $\qquad\square$

### E.4 FIND MODIFIED PAIRS

---
**Algorithm 6** Find modified pairs

---
1: **data structure** DYNAMICGEOSPAR           $\triangleright$ Theorem E.3
2:  **procedure** FINDMODIFIEDPAIRS$(T, \mathcal{P}, p, p')$     $\triangleright$ move $p$ to $p'$, Lemma E.8
3:   $l_p \leftarrow$ QTFASTPL$(T, p)$            $\triangleright$ Lemma B.18
4:   $l_{p'} \leftarrow$ QTFASTPL$(T, p')$           $\triangleright$ Lemma B.18
5:   $\mathcal{S} \leftarrow \emptyset$
6:   $\mathcal{P}^{\text{new}} \leftarrow \mathcal{P}$
7:   **for** $n \in$ quad tree nodes on the path from $l_p$ to the quad tree root **do**
8:    $P_n \leftarrow$ WFINDPAIRS$(\mathcal{P}, n)$         $\triangleright$ Lemma B.18
9:    **for** every pair $(A, B) \in P_n$ **do**
10:     **if** $A' = \{p\}$ or $B' = \{p\}$ **then**
11:      $A', B' \leftarrow \emptyset$
12:     **else**
13:      Remove $p$ from $(A, B)$ and obtain $(A', B')$
14:     **end if**
15:     $\mathcal{S} \leftarrow \mathcal{S} \cup \{(A, B, A', B')\}$
16:    **end for**
17:   **end for**
18:   **for** $n \in$ quad tree nodes on the path from $l_{p'}$ to the quad tree root **do**
19:    $P_n \leftarrow$ WFINDPAIRS$(\mathcal{P}, n)$         $\triangleright$ Lemma B.18
20:    **for** every pair $(A, B) \in P_n$ **do**
21:     Add $p'$ to $(A, B)$ and obtain $(A', B')$
22:     $\mathcal{S} \leftarrow \mathcal{S} \cup \{(A, B, A', B')\}$
23:    **end for**
24:   **end for**
25:   **for** $(A, B, A', B') \in \mathcal{S}$ **do**
26:    $\mathcal{P}^{\text{new}} \leftarrow$ replace $(A, B) \in \mathcal{P}$ with $(A', B')$.
27:   **end for**
28:   $T^{\text{new}} \leftarrow$ QTINSERTP(QTDELETEP$(T, p), p')$    $\triangleright$ Lemma B.18
29:   **return** $\mathcal{S}, T^{\text{new}}, \mathcal{P}^{\text{new}}$
30: **end procedure**
31: **end data structure**

---

WSPD is stored as a list of pairs $\mathcal{P}$ that supports:

- WFINDPAIRS$(\mathcal{P}, A)$, find all pairs $(A, B)$ and $(B, A) \in \mathcal{P}$ time linear in the output size.

**Lemma E.8.** *Given a compressed quad tree $T$ of a $O(k)$-dimensional point set $P$, a WSPD $\mathcal{P}$ computed from $T$, a point $p \in P$ and another point $p'$, in the output of Algorithm 6, $T^{\text{new}}$ is a quad tree $T^{\text{new}}$ of $P\backslash\{p\} \cup \{p'\}$, $\mathcal{P}^{\text{new}}$ is a WSPD of $P\backslash\{p\} \cup \{p'\}$ and $\mathcal{S}$ is a collection of tuples $(A, B, A', B')$. $\mathcal{P}^{\text{new}}$ can be obtained by doing the following:*

*For all $(A, B, A', B') \in \mathcal{S}$, replace $(A, B) \in \mathcal{P}$ with $(A', B')$.*

*This can be done in $2^{O(k)} \log \alpha$ time.*

*Proof.* We divide the proof into the following parts.

**Correctness** By Lemma B.18, we have that, the new generated tree $T^{\mathrm{new}}$ is a quad tree of $P \backslash \{p\} \cup \{p'\}$.

We now show that, after replacing $(A, B) \in \mathcal{P}$ with $(A', B')$ for all $(A, B, A', B')$ in $\mathcal{S}$ in Line 26, we get a WSPD of the updated point set.

First in Line 3 and Line 3, we find the path from the root to the leaf node containing $p$ and $p'$. Then in the following two for-loops (Line 7 and Line 18), we iteratively visit the nodes on the paths. In each iteration, we find the WS pairs related to the node by calling WFINDPAIRS. We record the original sets and the updated sets. Then in Line 26, we replace the original pairs by the updated pairs to get the up-to-date pair list.

**Running time** By Lemma B.18, the two calls to QTFASTPL takes $O(\log n)$ time. For each of $p$ and $p'$, there are at most $2^{O(k)} \log n$ pairs that can contain $p$ or $p'$. Therefore, the total running time of WFINDPAIRS is $O(2^{O(k)}) \log n$, and there are $2^{O(k)} \log n$ tuples in $\mathcal{S}$. The number of times that the loops on lines 9 and 20 are executed is at most $2^{O(k)} \log n$. Hence the total time complexity of FINDMODIFIEDPAIRS is $2^{O(k)} \log n$.

$\square$

### E.5 LINEAR TIME RESAMPLING ALGORITHM

Here in this section, we state our linear time resampling algorithm.

---

**Algorithm 7** Linear Time Resampling Algorithm

---

1: **procedure** RESAMPLE($E, A, B, A', B', s$)  $\quad\quad\quad\quad\quad\quad\quad\quad\quad\quad\quad$ ▷ Lemma E.9
2: $\quad$ $E \leftarrow E \cap (A' \times B')$
3: $\quad$ $\mathcal{R} \leftarrow \emptyset$
4: $\quad$ $q \leftarrow \frac{|(A \times B) \cap (A' \times B')|}{|A' \times B'|}$
5: $\quad$ **for** $j = 1 \to s$ **do**
6: $\quad\quad$ Draw a random number $x$ from $[0, 1]$
7: $\quad\quad$ **if** $x \leq q$ **then**
8: $\quad\quad\quad$ **if** $E \backslash \mathcal{R} \neq \emptyset$ **then**
9: $\quad\quad\quad\quad$ Sample one pair from $E$ (without repetition) and add it to $\mathcal{R}$
10: $\quad\quad\quad$ **else** $\quad\quad\quad\quad\quad\quad\quad\quad\quad\quad\quad\quad\quad\quad\quad\quad\quad\quad$ ▷ all points of $E$ are sampled
11: $\quad\quad\quad\quad$ Sample one pair from $((A \times B) \cap (A' \times B')) \backslash E$ (without repetition) and add it to $\mathcal{R}$
12: $\quad\quad\quad$ **end if**
13: $\quad\quad$ **else**
14: $\quad\quad\quad$ Sample one pair of points $(a, b)$ from $(A' \times B') \backslash (A \times B)$ and add it to $\mathcal{R}$
15: $\quad\quad$ **end if**
16: $\quad$ **end for**
17: $\quad$ **return** $\mathcal{R}$
18: **end procedure**

---

**Lemma E.9** (Resample). *Let $C_{\mathrm{j1}}$ be the constant defined in Lemma B.6. Let $V$ be a set, $A, B$ be subsets of $V$ such that $A \cap B = \emptyset$, $A', B'$ be two sets that are not necessarily subsets of $V$ such that*

$$A' \cap B' = \emptyset \text{ and } |(A \times B) \triangle (A' \times B')| < o(\frac{|A' \times B'|}{|A'| + |B'|}).$$

*Let $n = |V \cup A' \cup B'|$. Let $E$ be a subset of $V \times V$.*

*Let $H$ be a graph on vertex set $V$, $A, B \subset V$ and $A \cap B = \emptyset$. Let $A', B'$ be two other vertex sets such that $A' \cap B' = \emptyset$ ($A'$ and $B'$ do not have to be subsets of $V$). If*

- *$E$ is a uniform sample of size*

$$\epsilon^{-2} n^{C_{\mathrm{j1}} \cdot (L/k)} (|A| + |B|) \log(|A| + |B|)$$

*from $A \times B$.*

- $s = \epsilon^{-2} n^{C_{j1} \cdot (L/k)} (|A'| + |B'|) \log(|A'| + |B'|)$

- $|s - |E|| = n^{o(1)}$

*then with high probability,* RESAMPLE *generates a uniform sample of size $s$ from $A' \times B'$ in $n^{o(1)}$ time. Moreover, with probability at least $1 - \delta$, the size of difference between the new sample and $E$ is $n^{o(1)}$.*

*Proof.* To show that the sample is uniform, we can see this sampling process as follows: To draw $s$ samples from $A' \times B'$, the probability of each sample being drawn from $(A' \times B') \cap (A \times B)$ is

$$\frac{|(A' \times B') \cap (A \times B)|}{|A' \times B'|}.$$

Therefore, for each sample, with this probability, we draw this sample from $(A' \times B') \cap (A \times B)$ (line 7) and sample from $(A' \times B') \backslash (A \times B)$ otherwise (line 10).

Since $E$ is a uniform sample from $A \times B$, $E \cap (A' \times B')$ is a uniform sample from $(A \times B) \cap (A' \times B')$ and any uniformly randomly chosen subset of it is also a uniform sample from $(A \times B) \cap (A' \times B')$. Hence, to sample pairs from $(A \times B) \cap (A' \times B')$, we can sample from $E \cap (A \times B)$ first (line 9) and sample from outside $E \cap (A \times B)$ when all pairs in $E \cap (A \times B)$ are sampled (line 11). The resulting set is a uniform sample from $A' \times B'$.

To see the size difference between $E$ and $\mathcal{R}$, we note that since

$$|(A \times B) \triangle (A' \times B')| < o(\frac{|A' \times B'|}{|A'| + |B'|}),$$

the probability

$$\frac{|(A \times B) \cap (A' \times B')|}{|A' \times B'|} \geq 1 - \frac{|(A \times B) \triangle (A' \times B')|}{|A' \times B'|}$$

$$\geq 1 - o(\frac{1}{|A'| + |B'|})$$

Therefore, to draw $s = \epsilon^{-2} n^{C_{j1} \cdot (L/k)} (|A'| + |B'|) \log(|A'| + |B'|)$ samples from $A' \times B'$, the expectation of number of samples drawn from $(A' \times B') \backslash (A \times B)$ is at most

$$\epsilon^{-2} n^{C_{j1} \cdot (L/k)} (|A'| + |B'|) \log(|A'| + |B'|) \cdot o(\frac{1}{|A'| + |B'|}) \leq \epsilon^{-2} n^{C_{j1} \cdot (L/k)} \log(|A'| + |B'|)$$

By Markov inequality, with high probability $1 - \delta$, at most

$$\delta^{-1} \epsilon^{-2} n^{C_{j1} \cdot (L/k)} \log(|A'| + |B'|)$$

pairs were drawn from $(A' \times B') \backslash (A \times B)$.

Now we analyze the time complexity. The loop runs for $O(s)$ time and each sample can be done in constant time. Therefore, the total time complexity is $O(s)$. By the third bullet point, this is in worst case $O(n \log n)$ time. $\qquad \square$

### E.6 EFFICIENT SUBLINEAR TIME RESAMPLING ALGORITHM

Algorithm 7 returns a set of pairs that is with high probability close to the input set $E$. However, since it needs to sample all $s$ pairs, the time complexity is bad. We modify it by trying to remove samples from $E$ instead of adding pairs from $E$ to the new sample and obtain Algorithm 8.

**Lemma E.10** (Fast resample)**.** *Let $C_{j1}$ be the constant defined in Lemma B.6. Let $V$ be a set, $A, B$ be subsets of $V$ such that $A \cap B = \emptyset$, $A', B'$ be two sets that are not necessarily subsets of $V$ such that*

$$A' \cap B' = \emptyset \text{ and } |(A \times B) \triangle (A' \times B')| < o(\frac{|A' \times B'|}{|A'| + |B'|}).$$

*Let $n = |V \cup A' \cup B'|$. Let $E$ be a subset of $V \times V$. If*

---

**Algorithm 8** Efficient Sublinear Time Resampling Algorithm

---

1: **procedure** FASTRESAMPLE($E, A, B, A', B', s$)          ▷ Lemma E.10
2:      $\mathcal{R} \leftarrow \emptyset$
3:      $E \leftarrow E \cap (A' \times B')$
4:      Draw a random number $x$ from Binomial($s, \frac{|(A' \times B') \setminus (A \times B)|}{|A' \times B'|}$)
5:      Add to $\mathcal{R}$ $x$ points uniformly drawn from $(A' \times B') \setminus (A \times B)$ (without repetition)
6:      **if** $|E| > s - x$ **then**
7:          Draw $x + |E| - s$ pairs from $E$ uniformly randomly (without repetition)
8:          Add pairs in $E$ to $\mathcal{R}$ except these $x + |E| - s$ pairs drawn on line 7
9:      **else**
10:         Add all pairs in $E$ to $\mathcal{R}$
11:         Add to $\mathcal{R}$ $(s - x - |E|)$ points uniformly drawn from $(A' \times B') \setminus (A \times B) \setminus E$ (without repetition)
12:      **end if**
13:      **return** $\mathcal{R}$
14: **end procedure**

---

- $E$ is a uniform sample of size $\epsilon^{-2} n^{C_{\mathrm{j1}} \cdot (L/k)} (|A| + |B|) \log(|A| + |B|)$ from $A \times B$.

- $s = \epsilon^{-2} n^{C_{\mathrm{j1}} \cdot (L/k)} (|A'| + |B'|) \log(|A'| + |B'|)$

- $|E| < o(|A \times B|)$

- $s < o(|A' \times B'|)$

- $|s - |E|| = n^{o(1)}$

then with high probability, FASTRESAMPLE *generates a uniform sample of size $s$ from $A' \times B'$ in $n^{o(1)}$ time. Moreover, with probability at least $1 - \delta$, the size of difference between the new sample and $E$ is $n^{o(1)}$.*

*Proof.* To show that the sample is uniform, we can see this sampling process as follows: To draw $s$ samples from $A' \times B'$, the probability of each sample being drawn from $(A' \times B') \setminus (A \times B)$ is

$$\frac{|(A' \times B') \setminus (A \times B)|}{|A' \times B'|}.$$

Therefore, the number of samples drawn from $(A' \times B') \setminus (A \times B)$ satisfies a binomial distribution with parameters

$$s \text{ and } \frac{|(A' \times B') \setminus (A \times B)|}{|A' \times B'|}.$$

Let $x$ be such a binomial random variable, we sample $x$ pairs from $(A' \times B') \setminus (A \times B)$ and the rest from $(A' \times B') \cap (A \times B)$.

Since $E$ is a uniform sample from $A \times B$, $E \cap (A' \times B')$ is a uniform sample from $(A \times B) \cap (A' \times B')$, and any uniformly randomly chosen subset of it is also a uniform sample from $(A \times B) \cap (A' \times B')$. Hence, if $|E| > s - x$, we take $s - x$ pairs from $E$ by discarding $|E| - s + x$ pairs in $E$ (line 7 and 8). If $|E| \leq s - x$, we take all samples from $E$ and add $s - x - |E|$ pairs from $(A' \times B') \cap (A \times B)$ (line 11).

Now we try to bound the difference between $E$ and $\mathcal{R}$ and the time complexity. Since $x$ is drawn from a binomial distribution, we have that

$$\mathbb{E}[x] = s \cdot \frac{|(A' \times B') \setminus (A \times B)|}{|A' \times B'|}.$$

By Markov inequality,

$$\Pr\left[x > \frac{s}{\delta} \cdot \frac{|(A' \times B') \setminus (A \times B)|}{|A' \times B'|}\right] \leq \delta$$

Since $|(A \times B) \triangle (A' \times B')| < o(\frac{|A' \times B'|}{|A'| + |B'|})$, with probability at least $1 - \delta$,

$$x < \delta^{-1} \cdot \epsilon^{-2} \cdot n^{C_{\mathrm{j1}} \cdot (L/k)}(|A'| + |B'|) \log(|A'| + |B'|) \cdot o(\frac{1}{|A'| + |B'|})$$

$$< \delta^{-1} \epsilon^{-2} n^{C_{\mathrm{j1}} \cdot (L/k)} \log n$$

Therefore, drawing new samples (lines 7 and 8 or line 11) takes $O(x + |s - |E||)$ time. The difference between $E$ and the output sample set is also at most $O(x + |s - |E||) = n^{o(1)}$. With probability at least $1 - \delta$, we have that

$$x \leq \delta^{-1} \cdot \epsilon^{-2} \cdot n^{o(1)}.$$

Since $|s - |E|| \leq n^{o(1)}$, the overall time complexity and the difference between $E$ and the output set are at most $\delta^{-1} \cdot \epsilon^{-2} \cdot n^{o(1)}$.

Thus we complete the proof. $\qquad\square$

### E.7    A Data Structure That Can Handle $O(n)$ Updates

In this section, we combine the above algorithms and state our data structure.

---

**Algorithm 9** Data structure update

---

1: **data structure** DynamicGeoSpar $\qquad\qquad\qquad\qquad\qquad\qquad\qquad\qquad\qquad$ ▷ Theorem E.3
2: **procedure** Update($p \in \mathbb{R}^d, p' \in \mathbb{R}^d$) $\qquad\qquad\qquad\qquad\qquad\qquad\qquad\qquad$ ▷ Lemma E.11
3: $\qquad P^{\mathrm{new}} \leftarrow P \cup p' \backslash p$
4: $\qquad Q^{\mathrm{new}} \leftarrow Q \cup (\Pi p') \backslash (\Pi p)$ $\qquad\qquad$ ▷ $\Pi$ is a projection stored in memory and fixed over all the iterations
5: $\qquad \mathcal{S}, T^{\mathrm{new}}, \mathcal{P}^{\mathrm{new}} \leftarrow$ FindModifiedPairs$(T, P, \Pi p, \Pi p')$ $\qquad\qquad\qquad\qquad$ ▷ Algorithm 6
6: $\qquad \mathcal{H}^{\mathrm{new}} \leftarrow \mathcal{H}$
7: $\qquad$ **for** all $(A, B, A', B') \in \mathcal{S}$ **do**
8: $\qquad\qquad E \leftarrow$ Edges$(A, B)$
9: $\qquad\qquad$ Scale each edge in $E$ by $\epsilon^{-2}(n^{O(L/k)}(|A| + |B|) \log(|A| + |B|))/|A||B|$
10: $\qquad\qquad X, Y, X', Y' \leftarrow d$-dimensional points corresponding to $A, B, A', B'$
11: $\qquad\qquad$ **if** $|(A \times B) \triangle (A' \times B')| < o(\frac{|A' \times B'|}{|A'| + |B'|})$ **then**
12: $\qquad\qquad\qquad s \leftarrow \epsilon^{-2} n^{O(L/k)}(|X'| + |Y'|) \log(|X'| + |Y'|)$
13: $\qquad\qquad\qquad E^{\mathrm{new}} \leftarrow$ FastResample$(E, X, Y, X', Y', s)$ $\qquad\qquad\qquad\qquad$ ▷ Algorithm 8
14: $\qquad\qquad\qquad$ Scale each edge in $E^{\mathrm{new}}$ by $|X'||Y'|/s$
15: $\qquad\qquad$ **else**
16: $\qquad\qquad\qquad E^{\mathrm{new}} \leftarrow$ all edges in Biclique$(X', Y')$
17: $\qquad\qquad$ **end if**
18: $\qquad\qquad$ Edges.Update$(A, B, A', B', E, E^{\mathrm{new}})$ $\qquad\qquad$ ▷ Change $(A, B, E)$ to $(A', B', E^{\mathrm{new}})$
19: $\qquad\qquad \mathcal{H}^{\mathrm{new}} \leftarrow \mathcal{H}^{\mathrm{new}} \backslash E \cup E^{\mathrm{new}}$
20: $\qquad$ **end for**
21: $\qquad \mathcal{H} \leftarrow \mathcal{H}^{\mathrm{new}}$
22: $\qquad P \leftarrow P^{\mathrm{new}}$
23: $\qquad \mathcal{P} \leftarrow \mathcal{P}^{\mathrm{new}}$
24: $\qquad Q \leftarrow Q^{\mathrm{new}}$
25: $\qquad T \leftarrow T^{\mathrm{new}}$
26: **end procedure**
27: **end data structure**

---

**Lemma E.11.** *Given two points $p, p' \in \mathbb{R}^d$, with high probability $1 - \delta$, function* Update *(Algorithm 9) can handle $O(n)$ updates to the geometric graph and can update the $\epsilon$-spectral sparsifier in*

$$O(dk + \delta^{-1} \epsilon^{-2} n^{o(1)} \log \alpha)$$

*time per update. Moreover, after each update, the number of edge weight that are changed in the sparsifier is at most $\delta^{-1} \epsilon^{-2} n^{o(1)} \log \alpha$.*

*Proof.* Similar to Lemma E.7, in order for the updated $\mathcal{H}$ to be a spectral sparsifier of $\mathsf{K}_G$, we need

- After removing $\Pi p$ and adding $\Pi p'$, the resulting JL projection $Q$ still has distortion at most $n^{1/k}$.

- The WSPD of $Q$ is updated to a WSPD of $Q \backslash \{\Pi p\} \cup \{\Pi p'\}$

- For each WS pair $(A', B')$ in the new WSPD, let $X'$ and $Y'$ be $A'$ and $B'$'s corresponding $d$-dimensional point set respectively, we can obtain a uniform sample of

$$\epsilon^{-2}(n^{O(L/k)}(|X'| + |Y'|)\log(|X'| + |Y'|))$$

  edges from $\text{Biclique}(X', Y')$.

For each of the above requirement, we divide the proof into the following paragraphs.

**Bounding on the distortion of JL distance** To show the first requirement, we note that by Lemma B.6, if the JL projection matrix is initialized with $O(n)$ points, after at most $O(n)$ updates, with high probability, the distance distortion between two points is still bounded above by $n^{O(1/k)}$.

**Update to WSPD** To show the second requirement, FINDMODIFIEDPAIRS returns a collections $\mathcal{S}$ of pairs updates. By Lemma E.8, for each $(A, B, A', B') \in \mathcal{S}$, after replacing pair $(A, B) \in \mathcal{P}$ with pair $(A', B')$, we obtain an updated WSPD.

**Sample size guarantee** To show the third requirement, for each $(A, B, A', B')$ in $\mathcal{S}$, let $X, Y, X', Y'$ be their corresponding $d$-dimensional point sets. We resample

$$\epsilon^{-2}n^{O(L/k)}(|X'| + |Y'|)\log(|X'| + |Y'|)$$

edges from $\text{Biclique}(X', Y')$ by updating the edges sampled from $\text{Biclique}(X, Y)$. To do this, we first multiply each edge weight in $\text{EDGES}(X, Y)$ by

$$\epsilon^{-2}(n^{O(L/k)}(|X| + |Y|)\log(|X| + |Y|))/|X||Y|$$

so that each edge has the same weight in $E$ and in $\text{biclique}(X', Y')$. Then we apply FASTRESAMPLE. Since

- $|(X \times Y) \triangle (X' \times Y')| \leq o(\frac{|X' \times Y'|}{|X'| + |Y'|})$ (line 11)

- $E$ is a uniform sample from $X \times Y$ of size $\epsilon^{-2}n^{C_{j1} \cdot (L/k)}(|X| + |Y|)\log(|X| + |Y|)$ (line 8 and definition of EDGES)

- $s = \epsilon^{-2}n^{O(L/k)}(|X'| + |Y'|)\log(|X'| + |Y'|)$ (line 12)

- $|s - |E|| = O(\log n)$, because $||X| + |Y| - |X'| - |Y'||$ is at most 1.

by Lemma E.10, the new sample can be viewed as a uniform sample from $\text{biclique}(X', Y')$.

Similar to Lemma E.7, the edges uniformly sampled from $\text{Biclique}(X', Y')$ form a $\epsilon$-spectral sparsifier of $\text{Biclique}(X', Y')$ after scaling each edge weight by

$$|X'||Y'|/(\epsilon^{-2}(n^{O(L/k)}(|X'| + |Y'|)\log(|X'| + |Y'|))).$$

If the number of edges in $\text{Biclique}(X', Y')$ itself is

$$O\left(\epsilon^{-2}(n^{O(L/k)}(|X'| + |Y'|)\log(|X'| + |Y'|))\right),$$

we use all edges in the biclique without scaling. The union of all sampled edges remains a spectral sparsifier of $\mathsf{K}_G$.

The projection can be updated in $O(dk)$ time, where $d$ is the ambient dimension of the points and $k = o(\log n)$.

By Lemma E.8, FINDMODIFIEDPAIRS takes $O(2^{O(k)} \log(\alpha))$ time and the returned collection $\mathcal{S}$ contains at most $O(2^{O(k)} \log(\alpha))$ changed pairs.

By Lemma E.10, with high probability $1 - \delta$ resampling takes $\delta^{-1} \epsilon^{-2} n^{o(1)}$ time and the number of new edges in the sample is $\delta^{-1} \epsilon^{-2} n^{o(1)}$.

Therefore, with high probability, the total number of edge updates in $\mathcal{H}$ is with high probability

$$\delta^{-1} \epsilon^{-2} 2^{O(k)} \log \alpha n^{o(1)} = \delta^{-1} \epsilon^{-2} n^{o(1)} \log \alpha,$$

and the time needed to update the sparsifier is

$$O(dk + \epsilon^{-2} \delta^{-1} n^{o(1)} \log \alpha).$$

$\square$

### E.8 A DATA STRUCTURE THAT CAN HANDLE FULLY DYNAMIC UPDATE

By the limitation of the ultra low dimensional JL projection, when it needs to handle more than $O(n)$ projections, the $n^{C_{jl} \cdot (1/k)}$ distortion bound cannot be preserved with high probability. Therefore, Lemma E.11 states that DYNAMICGEOSPAR can only handle $O(n)$ updates.

This essentially gives us an online algorithm, with support of batch update. Under the setting of online batch, the dynamic data structure $\mathfrak{D}$ undergoes batch updates defined by these two parameters: the number of batches, denoted by $\zeta$, and the sensitivity parameter, denoted by $w$. $\mathfrak{D}$ has one initialization phase and $\zeta$ phases: an initialization phase and $\zeta$ update phases and in each update phase, the data structure $\mathfrak{D}$ receives updates for no more than $w$ times.

This algorithm is designed to maintain $\mathfrak{D}$ under the update batches. The data structure is maintained to exactly match the original graph after series of update batches. We define the *amortized randomized update time* $t$ to be the time such that, with every batch size less than $w$, the running time of each update to data structure is no more than $t$. The goal of this section is to minimize the time $t$. We first introduce the following useful lemma from literature, which introduces the framework of the online-batch setting.

**Lemma E.12** (Section 5, (Nanongkai et al., 2017)). *We define $G$ to be a geometric graph, with updates come in batches. Let $\zeta \in \mathbb{R}$ denote batch number. Let $w \in \mathbb{R}$ denote the sensitivity parameter. Then there exists a data structure $\mathfrak{D}$ with the batch number of $\zeta$ and sensitivity of $w$, which supports:*

- *An initialization procedure which runs in time $t_{\text{initialize}}$;*

- *An update procedure which runs in time $t_{\text{update}}$.*

*The two running time parameter $t_{\text{initialize}}$ and $t_{\text{update}}$ are defined to be functions such that, they send the maximum value of measures of the graph to non-negative numbers. For example, the upper bounds of the edges.*

*Then we have the result that, for any parameter $\xi$ such that $\xi \leq \min\{\zeta, \log_6(w/2)\}$, there exists a fully dynamic data structure consists of a size-$O(2^\xi)$ set of data structures $\mathfrak{D}$. It can initialize in time $O(2^\xi \cdot t_{\text{initialize}})$. And it has update time of $O\left(4^\xi \cdot \left(t_{\text{initialize}}/w + w^{(1/\xi)} t_{\text{update}}\right)\right)$ in the worst case. When the data structure is updated every time, the update procedure can select one instance from the set, which satisfies that*

1. *The selected instance of $\mathfrak{D}$ matches the updated graph.*

2. *The selected instance of $\mathfrak{D}$ has been updated for at most $\xi$ times, and the size of the update batch every time is at most $w$.*

By Lemma E.5, the initialization time of DYNAMICGEOSPAR is

$$O(ndk + \epsilon^{-2} n^{1+o(1)}) \log \alpha.$$

By Lemma E.11, the update time of DYNAMICGEOSPAR is

$$\delta^{-1} \epsilon^{-2} n^{o(1)} \log \alpha$$

per update and it can handle $O(n)$ batches of updates, each containing 1 update. Therefore, we can apply Lemma E.12 to DYNAMICGEOSPAR with $w = 1$ and $\zeta = O(n)$. We obtain a fully dynamic update data structure as stated below.

**Corollary E.13** (Corollary of Lemma E.5, E.11, and E.12). *There is a fully dynamic algorithm with initialization time $O(n^{1+(o(1))} \log \alpha)$ and update time $O(n^{o(1)})$.*

Corollary E.13 completes the proof of Theorem E.3.

## F  MAINTAINING A SKETCH OF AN APPROXIMATION TO MATRIX MULTIPLICATION

The goal of this section is to prove the following statement,

**Theorem F.1** (Formal version of Theorem 2.3). *Let $G$ be a $(C, L)$-lipschitz geometric graph on $n$ points. Let $v$ be a vector in $\mathbb{R}^d$. Let $k$ denote the sketch size. There exists an data structure MULTIPLY that maintains a vector $\widetilde{z}$ that is a low dimensional sketch of an $\epsilon$-approximation of the multiplication $L_G \cdot v$, where $b$ is said to be an $\epsilon$-approximation of $L_G x$ if*

$$\|b - L_G x\|_2 \leq \epsilon \|L_G\|_F \cdot \|x\|_2.$$

MULTIPLY *supports the following operations:*

- UPDATEG$(x_i, z)$: *move a point from $x_i$ to $z$ and thus changing $\mathsf{K}_G$. This takes $dk + n^{o(1)} \log \alpha$ time, where $\alpha$ is the aspect ratio of the graph.*

- UPDATEV$(\delta_v)$: *change $v$ to $v + \delta_v$. This takes $O(\log n)$ time.*

- QUERY$()$: *return the up-to-date sketch.*

We divide the section into the following parts. Section F.1 gives the high level overview of the section. Section F.2 introduces the necessity of sketching. Section F.3 introduces our algorithms.

### F.1  HIGH LEVEL OVERVIEW

The high level idea is to combine the spectral sparsifier defined in Section E and a sketch matrix to compute a sketch of the multiplication result and try to maintain this sketch when the graph and the vector change. We first revisit the definition of spectral sparsifiers. Let $G = (V, E)$ be a graph and $H = (V, E')$ be a $\epsilon$-spectral sparsifier of $G$. Suppose $|V| = n$. By definition, this means

$$(1 - \epsilon)L_G \preceq L_H \preceq (1 + \epsilon)L_G$$

**Lemma F.2.** *Let $G$ be a graph and $H$ be a $\epsilon$-spectral sparsifier of $G$. $L_H x$ is an $\epsilon$-approximation of $L_G x$*

*Proof.* By Definition B.26, $(1 - \epsilon)L_G \preceq L_H \preceq (1 + \epsilon)L_G$. Applying Proposition B.3, we get

$$\|L_H x - L_G x\|_{L_G^\dagger} \leq \epsilon \|L_G x\|_{L_G^\dagger}$$

which means $L_H x$ is an $\epsilon$-approximation of $L_G x$. $\qquad\square$

Thus, to maintain a sketch of an $\epsilon$-approximation of $L_G x$, it suffices to maintain a sketch of $L_H x$.

### F.2  NECESSITY OF SKETCHING

We here justify the decision of maintaining a sketch instead of the directly maintaining the multiplication result. Let the underlying geometry graph on $n$ vertices be $G$ and the vector be $v \in \mathbb{R}^n$. When a point is moved in the geometric graph, a column and a row are changed in $L_G$. Without loss of generality, we can assume the first row and first column are changed. When this happens, if the first entry of $v$ is not 0, all entries will change in the multiplication result. Therefore, it takes at least $O(n)$ time to update the multiplication result. In order to spend subpolynomial time to maintain the multiplication result, we need to reduce the dimension of vectors. Therefore, we use a sketch matrix to project vectors down to lower dimensions.

**Lemma F.3** ((Johnson & Lindenstrauss, 1984)). *Let $\epsilon \in (0, 0.1)$ denote an accuracy parameter. Let $\delta \in (0, 0.1)$ denote a failure probability. Let $X = \{x_1, \cdots, x_n\} \in \mathbb{R}^d$ denote a set of points. Let $\Phi \in \mathbb{R}^{m \times n}$ denote a randomized sketching matrix that, if $m = O(\epsilon^{-2} \log(n/\delta))$, with probability $1 - \delta$, we have: for all $x \in X$*

$$(1 - \epsilon) \cdot \|x\|_2 \le \|\Phi x\|_2 \le (1 + \epsilon) \cdot \|x\|_2.$$

We also have the following result by using the sparse embedding matrix:

**Lemma F.4.** *Let $\Psi \in \mathbb{R}^{m \times n}$ be a sparse embedding matrix (Definition C.1) with $m = O(\epsilon^{-2} \cdot \log(1/\delta))$. Then for a vector $v \in \mathbb{R}^n$ and a matrix $L \in \mathbb{R}^{n \times n}$, we have $\|(L\Psi^\top \Psi v) - (Lv)\|_2 \le \epsilon \cdot \|v\|_2 \cdot \|L\|_F$, with probability at least $1 - \delta$.*

*Proof.* By Lemma C.5, we have $\Psi$ satisfies the $(\epsilon, \delta, \log(1/\delta))$-JL moment property. Since $\log(1/\delta) \ge 2$ is trivial, then by Lemma C.4, and rescaling $\epsilon$ with a constant factor, we complete the proof. □

### F.3 ALGORITHMS

#### F.3.1 MODIFICATION TO DYNAMICGEOSPAR

For the applications in Section F and G, we add a member DIFF and methods GETDIFF and GET-LAPLACIAN to DYNAMICGEOSPAR and change methods INIT and Update to initialize and update DIFF (Algorithm 10).

---

**Algorithm 10** Interfaces for getting $\Delta L_H$ after $H$ changes

```
 1: data structure DYNAMICGEOSPAR                                    ▷ Lemma F.5
 2: members
 3:     diff                          ▷ the difference in the Laplacian after the graph is updated
 4: EndMembers
 5:
 6: procedure GETDIFF()
 7:     diffValue ← diff
 8:     diff ← {}
 9:     return diffValue
10: end procedure
11:
12: procedure GETLAPLACIAN()
13:     return The Laplacian matrix of ℋ
14: end procedure
15:
16: procedure INITIZLIZE(x_i, z)
17:     ...                                                ▷ Content in Algorithm 4
18:     diff ← {}
19: end procedure
20: procedure UPDATE(P)
21:     ...                                                ▷ Content in Algorithm 9
22:     for each EDGES update pair (E, E^new) do
23:         for each e in E\E^new do
24:             Add −e to diff
25:         end for
26:         for each e in E^new\E do
27:             Add e to diff
28:         end for
29:     end for
30: end procedure
```

---

**Lemma F.5.** *In data structure DYNAMICGEOSPAR, suppose GETDIFF (Algorithm 10) is called right after each UPDATE. The returned diff is a sparse matrix of size $O(n^{o(1)} \log \alpha)$.*

*Proof.* By Theorem E.3, in expectation each update introduces $O(n^{o(1)} \log \alpha)$ edge changes in the sparsifier. Therefore, after an updates, there are at most $O(n^{o(1)} \log \alpha)$ entries in diff. $\qquad\square$

### F.3.2 DYNAMIC SKETCH ALGORITHM

Here we propose the dynamic sketch algorithm as follows.

---

**Algorithm 11** Maintaining a sketch of an approximation to multiplication

1: **data structure** MULTIPLY                                            ▷ Theorem F.1
2: **members**
3:     DYNAMICGEOSPAR dgs                                               ▷ This is the sparsifier $H$
4:     $\Phi, \Psi \in \mathbb{R}^{m \times n}$: two independent sketching matrices
5:     $\widetilde{L} \in \mathbb{R}^{m \times m}$                                                    ▷ A sketch of $L_H$
6:     $\widetilde{v} \in \mathbb{R}^m$                                                              ▷ A sketch of $v$
7:     $\widetilde{z} \in \mathbb{R}^m$                                               ▷ A sketch of the multiplication result
8: **EndMembers**
9:
10: **procedure** INIT($x_1, \cdots, x_n \in \mathbb{R}^d, v \in \mathbb{R}^n$)
11:     Initialize $\Phi$ and $\Psi$
12:     dgs.INITIALIZE($x_1, \ldots, x_n$)
13:     $\widetilde{v} \leftarrow \Psi v$
14:     $\widetilde{L} \leftarrow \Phi \cdot \text{dgs.GETLAPLACIAN}() \cdot \Psi^\top$
15:     $\widetilde{z} \leftarrow \widetilde{L} \cdot \widetilde{v}$
16: **end procedure**
17:
18: **procedure** UPDATEG($x_i, z \in \mathbb{R}^d$)
19:     dgs.UPDATE($x_i, z$)
20:     $\Delta\widetilde{L} \leftarrow \Phi \cdot \text{dgs.GETDIFF}() \cdot \Psi^\top$
21:     $\widetilde{z} \leftarrow \widetilde{z} + \Delta\widetilde{L} \cdot \widetilde{v}$
22:     $\widetilde{L} \leftarrow \widetilde{L} + \Delta\widetilde{L}$
23: **end procedure**
24:
25: **procedure** UPDATEV($\Delta v \in \mathbb{R}^n$)                              ▷ $\Delta v$ is sparse
26:     $\Delta\widetilde{v} \leftarrow \Psi \cdot \Delta v$
27:     $\widetilde{z} \leftarrow \widetilde{z} + \widetilde{L} \cdot \Delta\widetilde{v}$
28: **end procedure**
29:
30: **procedure** QUERY
31:     **return** $\widetilde{z}$
32: **end procedure**

---

Here we give the correctness proof of Theorem F.1.

*Proof of Theorem F.1.* We divide the proof into correctness proof and running time proof as follows.

**Correctness**    By Lemma F.2, $L_H x$ is an $\epsilon$-approximation of $L_G x$. It suffices to show that MULTIPLY maintains a sketch of $L_H x$.

In function INIT, a $\epsilon$-spectral sparsifier $H$ of $G$ is initialized on line 12. On line 14, $\widetilde{L}$ is computed as $\Phi L_H \Psi^\top$. Therefore, $\widetilde{z} = \Phi L_H \Psi^\top \Psi v$. By Lemma F.4 we have that

$$\|L_H \Psi^\top \Psi v - L_H v\|_2 \leq \epsilon \cdot \|L_H\|_F \cdot \|v\|_2. \tag{1}$$

And by Lemma F.3 one has

$$\|\widetilde{z}\|_2 \in (1 \pm \epsilon) \cdot \|L_H \Psi^\top \Psi v\|_2. \tag{2}$$

Then by Eq (1) and Eq (2) and rescaling $\epsilon$ we have that $\|\widetilde{z} - L_H x\|_2 \leq \epsilon \cdot \|L_H\|_F \cdot \|v\|_2$.

In function UPDATEG, the algorithms updates the spectral sparsifier (line 19) and obtains the difference in the Laplacian (line 20). Note that $\Delta \widetilde{L} \cdot \widetilde{v} = \Phi \Delta L_H \cdot \Psi^\top \cdot \Psi v$. Again, since in expectation $\Psi^\top \Psi = I_{n \times n}$ and $\Phi$ and $\Psi$ are chosen independently, in expectation $\Delta \widetilde{L} \cdot \widetilde{v} = \Phi \Delta L_H v$. Therefore, $\widetilde{z} + \Delta \widetilde{L} \cdot \widetilde{v} = \Phi(L_H + \Delta L_H)v$ is the updated sketch of $L_H x$.

**Running time**   By Theorem E.3, line 19 takes $O(dk + n^{o(1)} \log \alpha)$ time. By Lemma F.5, $\Delta \widetilde{L}$ is sparse with $\epsilon^{-2} n^{o(1)} \log \alpha$ non-zero entries. This implies line 20 takes $O(\epsilon^{-2} n^{o(1)} \log \alpha)$ time. So the overall time complexity of UPDATEG is

$$dk + \epsilon^{-2} n^{o(1)} \log \alpha.$$

In function UPDATEV, note that

$$\widetilde{L} \cdot \Delta \widetilde{v} = \Phi \cdot L_H \cdot \Psi^\top \cdot \Psi \Delta v.$$

Again, since in expectation $\Psi^\top \Psi = I_{n \times n}$ and $\Phi$ and $\Psi$ are chosen independently, in expectation

$$\widetilde{L} \cdot \Delta \widetilde{v} = \Phi L_H \Delta v.$$

Therefore,

$$\widetilde{z} + \widetilde{L} \cdot \Delta \widetilde{v} = \Phi L_H (v + \Delta v)$$

is the updated sketch of $L_H x$.

Since $\Delta v$ is sparse, $\Psi \Delta v$ can be computed in $O(m)$ time, and $\widetilde{z}$ can also be updated in $O(m)$ time, where $m = O(\epsilon^{-2} \log(n/\delta))$.

Since $\widetilde{z}$ is always an up-to-date sketch of $L_H \cdot v$, QUERY always returns a sketch of an approximation to $L_G x$ in constant time.

Thus we complete the proof. $\square$

# G   MAINTAINING A SKETCH OF AN APPROXIMATION TO SOLVING LAPLACIAN SYSTEM

In this section, we provide a data structure which maintans a sketch of an approximation to sovling Laplacian system. In other words, we prove the following theorem,

**Theorem G.1** (Formal version of Theorem 2.4). *Let $\mathsf{K}_G$ be a $(C, L)$-lipschitz geometric graph on $n$ points. Let $b$ be a vector in $\mathbb{R}^d$. There exists a data structure SOLVE that maintains a vector $\widetilde{z}$ that is a low dimensional sketch of multiplication $L_G^\dagger \cdot b$, where $\widetilde{z}$ is said to be an $\epsilon$-approximation of $L_G^\dagger b$ if*

$$\|\widetilde{z} - L_G^\dagger b\|_2 \le \epsilon \cdot \|L_G^\dagger\|_F \cdot \|b\|_2.$$

SOLVE *supports the following operations:*

- UPDATEG$(x_i, z)$: *move a point from $x_i$ to $z$ and thus changing $\mathsf{K}_G$. This takes $n^{o(1)}$ time.*

- UPDATEB$(\delta_b)$: *change $b$ to $b + \delta_b$. This takes $n^{o(1)}$ time.*

- QUERY$()$: *return the up-to-date sketch. This takes $O(1)$ time.*

By Fact B.3, for any vector $b$, $L_H^\dagger b$ is a $\epsilon$-spectral sparsifier of $L_G^\dagger b$. It suffices to maintain a sketch of $L_H b$.

When trying to maintain a sketch of a solution to $L_H x = b$, the classical way of doing this is to maintain $\overline{x}$ such that $\Phi L_H \overline{x} = \Phi b$. However, here $\overline{x}$ is still an $n$-dimensional vector and we want to maintain a sketch of lower dimension. Therefore, we apply another sketch $\Psi$ to $\overline{x}$ and maintain $\widetilde{x}$ such that $\Phi L_H \Psi^\top \widetilde{x} = \Phi b$.

*Proof of Theorem G.1.* We divide the proof into the following paragraphs.

**Analysis of INIT**  In function INIT, a $\epsilon$-spectral sparsifier $H$ of $G$ is initialized on line 13. On line 16, $\widetilde{L}^\dagger$ is computed as $(\Phi L_H \Psi^\top)^\dagger$. Therefore,

$$\widetilde{z} = (\Phi L_H \Psi^\top)^\dagger \Phi b,$$

which is a sketch of $L_H^\dagger x$. Thus, after INIT, $L_H x$, which is a sketch of an approximation to $L_G x$ is stored in $\widetilde{z}$.

**Analysis of UPDATEG**  In function UPDATEG, the algorithms updates the spectral sparsifier (line 21) and obtains the new Laplacian (line 22) and its pseudoinverse (line 23). Note that in line 24

$$\widetilde{L}_{\text{new}}^\dagger \cdot \widetilde{b} = (\Phi(L_H + \Delta L_H)\Psi^\top)^\dagger \Phi b$$

This is a sketch of $(L_H + \Delta L_H)^\dagger b$.

By Theorem E.3, line 21 takes $O(dk + n^{o(1)}\log\alpha)$ time.  By Lemma F.5, $\Delta\widetilde{L}$ is sparse with $\epsilon^{-2}n^{o(1)}\log\alpha$ non-zero entries.  This implies line 22 takes $O(\epsilon^{-2}n^{o(1)}\log\alpha)$ time.  Since $\widetilde{L}^\dagger$ is a $m \times m$ matrix and $m = O(\epsilon^{-2}\log(n/\delta))$, computing its pseudoinverse takes at most $m^\omega = (\epsilon^{-2}\log(n/\delta))^\omega$[6] time.

So the overall time complexity of UPDATEG is

$$O(dk) + \epsilon^{-2}n^{o(1)}\log\alpha + O((\epsilon^{-2}\log(n/\delta))^\omega).$$

**Analysis of UPDATEB**  In function UPDATEB, note that

$$\widetilde{L}^\dagger \cdot \Delta\widetilde{b} = (\Phi \cdot L_H \cdot \Psi^\top)^\dagger \cdot \Phi\Delta b.$$

Therefore, it holds that

$$\widetilde{z} + \widetilde{L}^\dagger \cdot \Delta\widetilde{b} = (\Phi \cdot L_H \cdot \Psi^\top)^\dagger \cdot \Phi(b + \Delta b).$$

This is a sketch of $L_H^\dagger(b + \Delta b)$.

Since $\Delta b$ is sparse, $\Phi\Delta b$ can be computed in $O(m)$ time, and $\widetilde{z}$ can also be updated in $O(m)$ time, where $m = O(\epsilon^{-2}\log(n/\delta))$.

**Analysis of QUERY**  Since $\widetilde{z}$ is always an up-to-date sketch of $L_H \cdot v$, QUERY always returns a sketch of an approximation to $L_G x$ in constant time.

Thus we complete the proof. $\qquad\qquad\square$

# H  DYNAMIC DATA STRUCTURE

In this section, we describe our data structure in Algorithm 12 to solve the dynamic distance estimation problem with robustness to adversarial queries. We need to initialize a sketch $\Pi \in \mathbb{R}^{k \times d}$ defined in Definition H.1, where $k = \Theta(\sqrt{\log n})$, and use the ultra-low dimensional projection matrix to maintain a set of projected points $\{\widetilde{x}_i \in \mathbb{R}^k\}_{i=1}^n$. During QUERY, the data structure compute the estimated distance between the query point $q \in \mathbb{R}^d$ and the data point $x_i$ by $n^{1/k} \cdot \sqrt{d/k} \cdot \|\widetilde{x}_i - \Pi q\|_2$.

## H.1  MAIN RESULT

In this section, we introduce our main results, we start with defining ultra-low dimensional JL matrix.

**Definition H.1** (Ultra-Low Dimensional JL matrix). *Let $\Pi \in \mathbb{R}^{k \times d}$ denote a random JL matrix where each entry is i.i.d. Gaussian.*

Next, we present our main result in accuracy-efficiency trade-offs, which relates to the energy consumption in practice.

---

[6]$\omega$ is the matrix multiplication constant

---

**Algorithm 12** Data Structure for Ultra-Low JL Distance Estimation

---

1: **data structure** ULTRAJL
2: **members**
3:     $d, n \in \mathbb{N}_+$                     ▷ $n$ is dataset size, $d$ is the data dimension
4:     $X = \{x_i \in \mathbb{R}^d\}_{i=1}^n$                     ▷ Set of points being queried
5:     $\widetilde{X} = \{\widetilde{x}_i \in \mathbb{R}^k\}_{i=1}^n$                 ▷ Set of projected points being queried
6:     $\epsilon \in (0, 0.1)$
7:     $k \in \mathbb{N}_+$                          ▷ $k$ is the dimension we project to
8:     $\Pi \in \mathbb{R}^{k \times d}$                        ▷ Definition H.1
9: **end members**
10:
11: **procedure** INIT($\{x_1, \cdots, x_n\} \subset \mathbb{R}^d, n \in \mathbb{N}_+, d \in \mathbb{N}_+, \epsilon \in (0, 0.1)$)     ▷ Lemma H.3, H.6
12:                                     ▷ We require that $d = \Theta(\log n)$
13:     $n \leftarrow n, d \leftarrow d, \delta \leftarrow \delta, \epsilon \leftarrow \epsilon$
14:     $k \leftarrow \Theta(\sqrt{\log n})$
15:     **for** $i = 1 \rightarrow n$ **do**
16:         $x_i \leftarrow x_i$
17:     **end for**
18:     **for** $i = 1 \rightarrow n$ **do**
19:         $\widetilde{x}_i \leftarrow \Pi \cdot x_i$
20:     **end for**
21: **end procedure**
22:
23: **procedure** UPDATE($i \in [n], z \in \mathbb{R}^d$)                    ▷ Lemma H.4, H.7
24:     $x_i \leftarrow z$
25:     $\widetilde{x}_i \leftarrow \Pi \cdot z$
26: **end procedure**
27:
28: **procedure** QUERY($q \in \mathbb{R}^d$)                         ▷ Lemma H.5, H.8
29:     **for** $i \in [n]$ **do**
30:         $u_i \leftarrow n^{1/k} \cdot \sqrt{d/k} \cdot \|\widetilde{x}_i - \Pi q\|_2$
31:     **end for**
32:     **return** $u$                                     ▷ $u \in \mathbb{R}^n$
33: **end procedure**
34: **end data structure**

---

**Theorem H.2** (Main result). *Let $d = \Theta(\log n)$. Let $k = \Theta(\sqrt{\log n})$. There is a data structure (Algorithm 12) for the Online Approximate Dynamic Ultra-Low Dimensional Distance Estimation Problem with the following procedures:*

- INIT($\{x_1, x_2, \ldots, x_n\} \subset \mathbb{R}^d, n \in \mathbb{N}_+, d \in \mathbb{N}_+, \epsilon \in (0, 1)$): *Given $n$ data points $\{x_1, x_2, \ldots, x_n\} \subset \mathbb{R}^d$, an accuracy parameter $\epsilon$, and input dimension $d$ and number of input points $n$ as input, the data structure preprocesses in time $O(ndk)$.*

- UPDATE($z \in \mathbb{R}^d, i \in [n]$): *Given an update vector $z \in \mathbb{R}^d$ and index $i \in [n]$, the UPDATEX takes $z$ and $i$ as input and updates the data structure with the new $i$-th data point in $O(dk)$ time.*

- QUERY($q \in \mathbb{R}^d$): *Given a query point $q \in \mathbb{R}^d$, the QUERY operation takes $q$ as input and approximately estimates the norm distances from $q$ to all the data points $\{x_1, x_2, \ldots, x_n\} \subset \mathbb{R}^d$ in time $O(nk)$ i.e. it outputs a vector $u \in \mathbb{R}^n$ such that:*

$$\forall i \in [n], \|q - x_i\|_2 \le u_i \le n^{O(1/k)} \cdot \|q - x_i\|_2$$

*with probability at least $1 - 1/\operatorname{poly}(n)$, even for a sequence of adversarially chosen queries.*

## H.2 TIME

In the section, we will provide lemmas for the time complexity of each operation in our data structure.

**Lemma H.3** (INIT time). *There is a procedure* INIT *which takes a set of d-dimensional vectors* $\{x_1, \cdots, x_n\}$, *a precision parameter* $\epsilon \in (0, 0.1)$ *and* $d, n \in \mathbb{N}_+$ *as input, and runs in* $O(ndk)$ *time.*

*Proof.* Storing every vector $x_i$ takes $O(nd)$ time. Computing and storing $\widetilde{x}_i$ takes $O(n \times dk) = O(ndk)$ time. Thus procedure INIT runs in $O(ndk)$ time. $\square$

We prove the time complexity of UPDATE operation in the following lemma:

**Lemma H.4** (UPDATE time). *There is a procedure* UPDATE *which takes an index* $i \in [n]$ *and a d-dimensional vector z as input, and runs in* $O(nk)$ *time.*

*Proof.* Updating $x_i$ takes $O(d)$ time. Update $\widetilde{x}_i$ takes $O(dk)$ time. Thus procedure UPDATE runs in $O(dk)$ time. $\square$

We prove the time complexity of QUERY operation in the following lemma:

**Lemma H.5** (QUERY time). *There is a procedure* QUERY *which takes a d-dimensional vector q as input, and runs in* $O(nk)$ *time.*

*Proof.* Computing $\Pi q$ takes $O(dk)$ time. Computing all the $u_i$ takes $O(nk)$ time. Thus procedure QUERY runs in $O(nk)$ time. $\square$

## H.3 CORRECTNESS

In this section, we provide lemmas to prove the correctness of operations in our data structure.

**Lemma H.6** (INIT correctness). *There is a procedure* INIT *which takes a set* $\{x_1, \cdots, x_n\}$ *of d-dimensional vectors and a precision parameter* $\epsilon \in (0, 0.1)$, *and stores an adjoint vector* $\widetilde{x}_i$ *for each* $x_i$.

*Proof.* During INIT operation in Algorithm 12, the data structure stores a set of adjoint vectors $\widetilde{x}_i \leftarrow \Pi \cdot x_i$ for $i \in [n]$. This completes the proof.

$\square$

Then we prove the correctness of UPDATE operation in Lemma H.7.

**Lemma H.7** (UPDATE correctness). *There is a procedure* UPDATE *which takes an index* $i \in [n]$ *and a d-dimensional vector z, and uses z to replace the current* $x_i$.

*Proof.* During UPDATE operation in Algorithm 12, the data structure update the $i$-th adjoint vector $\widetilde{x}_i$ by $\Pi \cdot z$. This completes the proof.

$\square$

We prove the correctness of QUERY operation in Lemma H.8.

**Lemma H.8** (QUERY correctness). *There is a procedure* QUERY *which takes a d-dimensional vector q as input, and output an n-dimensional vector u such that for each* $i \in [n]$, $\|q - x_i\|_2 \le u_i \le n^{O(1/k)} \cdot \|q - x_i\|_2$ *with probability* $1 - 2/n$.

*Proof.* The proof follows by Lemma H.9, Lemma I.2 and Lemma I.4. This completes the proof.

$\square$

### H.4  HIGH PROBABILITY

With Lemma B.9 and Lemma B.8 ready, we want to prove the following lemma:

**Lemma H.9** (High probability for each point). *For any integer $n$, let $d = c_0 \log n$. Let $k$ be a positive integer such that $k = \sqrt{\log n}$. Let $f$ be a map $f : \mathbb{R}^d \to \mathbb{R}^k$. Let $\delta_1 = n^{-c}$ denote the failure probability where $c > 1$ is a large constant. Let $c_0 > 1$ denote some fixed constant. Then for each fixed points $u, v \in \mathbb{R}^d$, such that,*

$$\|u - v\|_2^2 \leq \|f(u) - f(v)\|_2^2 \leq \exp(c_0 \cdot \sqrt{\log n})\|u - v\|_2^2.$$

*with probability $1 - \delta_1$.*

*Proof.* If $d \leq k$, the theorem is trivial. Else let $v', u' \in \mathbb{R}^k$ be the projection of point $v, u \in \mathbb{R}^d$ into $\mathbb{R}^k$. Then, setting $L = \|u' - v'\|_2^2$ and $\mu = \frac{k}{d}\|u - v\|_2^2$. We have that

$$
\begin{aligned}
\Pr[L \leq (n^{-2c/k}/e)\mu] &= \Pr[L \leq (n^{-2c/\sqrt{\log n}}/e)\mu] \\
&= \Pr[L \leq e^{-2c\sqrt{\log n}-1}\mu] \\
&\leq n^{-c}
\end{aligned}
\tag{3}
$$

where the first step comes from $k = \sqrt{\log n}$, the second step comes from $n^{-2e/\log n} = \exp(-2c\sqrt{\log n})$, and the third step comes from Lemma B.9.

By Lemma B.8, we have:

$$
\begin{aligned}
\Pr[L \geq n^{1/k}\mu] &= \Pr[L \geq n^{1/\sqrt{\log n}}\mu] \\
&= \Pr[L \geq \exp(c_0\sqrt{\log n})\mu] \\
&\leq \exp(-\log^{1.9} n) \\
&\leq n^{-c}
\end{aligned}
\tag{4}
$$

where the first step comes from the definition of $k = \sqrt{\log n}$, the second step follows that $n^{1/\sqrt{\log n}} = \exp(c_0\sqrt{\log n})$, the third step comes from Lemma B.8, and the fourth step follows that $\log^{1.9}(n)$ is bigger than any constant $c$.

Therefore, rescaling from Eq. (3) and Eq. (4) we have:

$$\|u - v\|_2^2 \leq \|f(u) - f(v)\|_2^2 \leq \exp(c_0 \cdot \sqrt{\log n})\|u - v\|_2^2.$$

with probability $1 - \delta_1$ where $\delta_1 = n^{-c}$.

$\square$

## I  SPARSIFIER IN ADVERSARIAL SETTING

In Section H, we get a dynamic distance estimation data structure with robustness to adversarial queries. Here in this section, we provide the analysis to generalize our spectral sparsifier to adversarial setting, including discussion on the aspect ratio $\alpha$ (Definition B.10).

### I.1  DISTANCE ESTIMATION FOR ADVERSARIAL SPARSIFIER

**Fact I.1.** *Let $\alpha$ be defined as Definition B.10. Let $N$ denote a $\epsilon_1$-net on the $\ell_2$ unit ball $\{x \in \mathbb{R}^d \mid \|x\|_2 \leq 1\}$, where $d = O(\log n)$ and $\epsilon_1 = O(\alpha^{-1})$. Then we have that $|N| \leq \alpha^{O(\log n)}$.*

**Lemma I.2.** *Let $\alpha$ be defined as Definition B.10. For any integer $n$, let $d = O(\log n)$, let $k = \sqrt{\log n}$. Let $f : \mathbb{R}^d \to \mathbb{R}^k$ be a map. If $\alpha = O(1)$, then for an $\epsilon_1$-net $N$ with $|N| \leq \alpha^{O(\log n)}$, for all $u, v \in N$,*

$$\|u - v\|_2^2 \leq \|f(u) - f(v)\|_2^2 \leq \exp(\sqrt{\log n})\|u - v\|_2^2$$

*with probability $1 - \delta_2$.*

*Proof.* By Lemma H.9, we have that for any fix set $V$ of $n$ points in $\mathbb{R}^d$, there exists a map $f : \mathbb{R}^d \to \mathbb{R}^k$ such that for all $u, v \in V$,

$$\|u - v\|_2^2 \le \|f(u) - f(v)\|_2^2 \le \exp(\sqrt{\log n})\|u - v\|_2^2,$$

with probability $1 - \delta_1$, where $\delta_1 = n^{-c}$.

We apply the lemma on $N$, and by union bound over the points in $N$, we have that for all points $u, v \in N$,

$$\|u - v\|_2^2 \le \|f(u) - f(v)\|_2^2 \le \exp(\sqrt{\log n})\|u - v\|_2^2,$$

with probability $1 - \delta_2$, where it holds that

$$
\begin{aligned}
\delta_2 &= \delta_1 \cdot |N|^2 \\
&\le \delta_1 \cdot \alpha^{O(\log n)} \\
&= n^{-c} \cdot \alpha^{O(\log n)} \\
&\le n^{O(1)-c},
\end{aligned}
$$

where the first step follows from union bound, the second step follows from $|N| \le \alpha^{O(\log n)}$, the third steps follows from $\delta_1 = n^{-c}$ (Lemma H.9), and the last step follows from $\alpha = O(1)$.

By choosing $c$ as a constant large enough, we can get the low failure probability. $\square$

**Corollary I.3** (Failure probability on $\alpha$ and $d$). *We have that, the failure probability of Lemma I.2 is bounded as long as $\alpha^d = O(\text{poly}(n))$.*

**Lemma I.4** (Adversarial Distance Estimation of the Spectral Sparsifier). *Let $k = \sqrt{\log n}$ be the JL dimension, $f : \mathbb{R}^d \to \mathbb{R}^k$ be a JL function. Let $\alpha$ be defined as Definition B.10. Then we have that for all points $u, v$ in the $\ell_2$ unit ball, there exists a point pair $u', v'$ which is the closest to $u, v$ respectively such that,*

$$\|u - v\|_2^2 \le \|f(u') - f(v')\|_2^2 \le \exp(\sqrt{\log n})\|u - v\|_2^2$$

*with probability $1 - n^{-c_1}$.*

*Proof.* By Lemma I.2, we have that for all $u', v' \in N$, it holds that

$$\|u' - v'\|_2^2 \le \|f(u') - f(v')\|_2^2 \le \exp(\sqrt{\log n})\|u' - v'\|_2^2 \tag{5}$$

with probability at least $1 - \delta_2$. From now on, we condition on the above event happens. Then for arbitrary $u, v \notin N$, there exists $u', v' \in N$ such that

$$\|u - u'\| \le \epsilon_1 \text{ and } \|v - v'\| \le \epsilon_1.$$

Recall that we set $\epsilon_1 = O(\alpha^{-1})$ and now all points are in $\ell_2$ ball, thus we have that $u' \ne v'$ for $u$ $\not v$. Then by triangle inequality we have that

$$
\begin{aligned}
\|u' - v'\|_2 &= \|u' - u + u - v + v - v'\|_2 \\
&\le \|u' - u\|_2 + \|u - v\|_2 + \|v - v'\|_2 \\
&\le \|u - v\|_2 + 2\epsilon_1 \\
&\le (1 + O(1)) \cdot \|u - v\|_2, \tag{6}
\end{aligned}
$$

where the last step follows by setting $\epsilon_1 = O(\|u - v\|_2) = O(\alpha^{-1})$. Similarly, we also have

$$\|u' - v'\|_2 \ge (1 - O(1)) \cdot \|u - v\|_2. \tag{7}$$

By the linearity of $f$ together with Eq.(5), (7) and (6), we have that

$$(1 - O(1)) \cdot \|u - v\|_2^2 \le \|f(u') - f(v')\|_2^2 \le (1 + O(1)) \cdot \exp(\sqrt{\log n})\|u - v\|_2^2.$$

Rescaling it, we get the desired result. Thus we complete the proof. $\square$

## I.2   SPARSIFIER IN ADVERSARIAL SETTING

Here in this section, we provide our result of spectral sparsifier that can handle adversarial updates.

**Theorem I.5** (Sparsifier in adversarial setting, formal version of Theorem 2.2). *Let $\alpha$ be the aspect ratio of a $d$-dimensional point set $P$ defined above. Let $k = O(\sqrt{\log n})$. If $\alpha^d = O(\mathrm{poly}(n))$, then there exists a data structure* DYNAMICGEOSPAR *that maintains a $\epsilon$-spectral sparsifier of size $O(n^{1+o(1)})$ for a $(C, L)$-Lipschitz geometric graph such that*

- DYNAMICGEOSPAR *can be initialized in*

$$O(ndk + \epsilon^{-2}n^{1+o(L/k)} \log n \log \alpha)$$

   *time.*

- DYNAMICGEOSPAR *can handle* adversarial *point location changes. For each change in point location, the spectral sparsifier can be updated in*

$$O(dk + 2^{O(k)}\epsilon^{-2}n^{o(1)} \log \alpha)$$

   *time. With high probability, the number of edges changed in the sparsifier is at most*

$$\epsilon^{-2}2^{O(k)}n^{o(1)} \log \alpha.$$

*Proof.* By Lemma I.4, the estimation data structure works for adversarial query points. Then the theorem follows by Lemma E.5, E.11, and E.12. □

## J   FIGURES

We list our figures here.

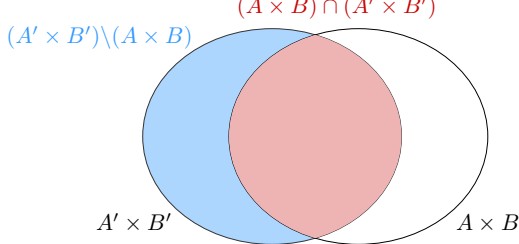

Figure 1: Division of the new biclique $(A' \times B')$: divided it into two parts (Blue part: $(A' \times B')\backslash(A \times B)$ and red part $(A \times B) \cap (A' \times B')$). And we sample from them respectively.

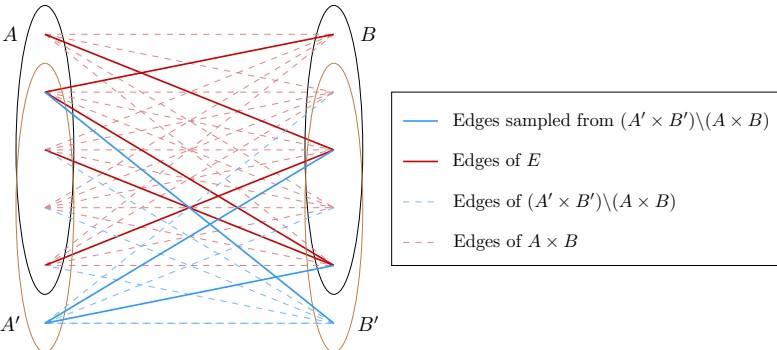

Figure 2: Resampling the biclique: $E$ (The red edges) is uniformly sampled from Biclique$(A, B)$. After $A \times B$ becomes $A' \times B'$, we resample from $E \cap (A' \times B')$ with specific probabilities.

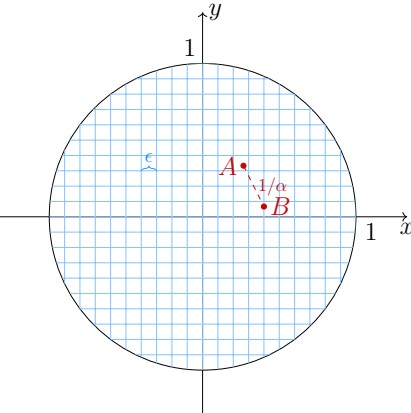

Figure 3: The net argument in our problem. Let $d = 2$. Here we restrict all the points to be in the $\ell_2$ unit ball. By the definition of aspect ratio $\alpha$, we know the minimum distance between two points is $1/\alpha$ ($A$ and $B$ in the figure). Thus, by setting $\epsilon = C \cdot \alpha^{-1}$ for some constant $C$ small enough, every pair of points is distinguishable.

## LLM USAGE DISCLOSURE

LLMs were used only to polish language, such as grammar and wording. These models did not contribute to idea creation or writing, and the authors take full responsibility for this paper's content.

