# OpenReview forum: "Dynamic Kernel Graph Sparsifiers"
_ICLR.cc/2026/Conference — ICLR 2026 Conference Desk Rejected Submission_

### Official Review · Reviewer_rC5G · 2025-10-29

**Soundness:** 3
**Presentation:** 3
**Contribution:** 3
**Rating:** 8
**Confidence:** 2

**Summary:**

This paper presents a fully-dynamic data structure to maintain an $\epsilon$-spectral sparsifier for a geometric graph. A geometric graph is defined here as a complete graph on a set of $n$ points $P \subset \mathbb{R}^d$, where edge weights are given by a kernel function $K(x_i, x_j)$. The dynamic operation considered is the update of a single point's location. This operation is challenging because it changes $O(n)$ edge weights simultaneously, rendering traditional dynamic graph algorithms (based on single edge updates) inefficient.

The primary contribution is a data structure, DYNAMICGEOSPAR, that achieves a subpolynomial ($n^{o(1)}$) update time and a nearly-linear ($n^{1+o(1)}$) initialization time. The core technical innovation is a new dynamic well-separated pair decomposition (WSPD). The method works by projecting the $d$-dimensional points into an ultra-low $k=o(\log n)$ dimension using a Johnson-Lindenstrauss (JL) transform. It then maintains a dynamic WSPD (using a compressed quadtree) in this low-dimensional space and uses a novel, efficient resampling technique to update the sparse set of edges corresponding to the original $d$-dimensional bicliques.

The paper further provides algorithms for maintaining randomized sketches for two related problems, also in $n^{o(1)}$ update time:
1.  Approximate matrix-vector multiplication with the graph Laplacian ($L_G v$).
2.  Approximate Laplacian solving ($L_G^\dagger b$).

Finally, the authors show that under a specific assumption balancing the data's aspect ratio ($\alpha$) and ambient dimension ($d$) such that $\alpha^d = O(poly(n))$, the data structure can be made robust against adaptive adversaries.

**Strengths:**

The work addresses a problem—fully-dynamic spectral sparsification for geometric graphs under *point location changes*—for which the authors claim no non-trivial algorithms were previously known. The primary technical contribution, a novel dynamic WSPD data structure combined with a "smooth resampling technique", seems original.

Kernel methods are a cornerstone of machine learning, and this paper provides a foundational tool for making them dynamic. The ability to efficiently maintain kernel-based graph structures under data changes is crucial for iterative optimization algorithms and real-world systems with streaming or evolving data. This paper successfully extends the important static results from (Alman et al., 2020) to the much more challenging dynamic setting.

**Weaknesses:**

- The result for robustness against adaptive adversaries hinges on the assumption that $\alpha^d = O(poly(n))$, where $\alpha$ is the aspect ratio and $d$ is the ambient dimension. This assumption appears very restrictive. For any moderately high-dimensional data (e.g., $d > \log n$), it requires the aspect ratio $\alpha$ to be extremely close to 1. This significantly limits the applicability of the adversarial-robust result to real-world, high-dimensional ML datasets, which are often not so well-behaved.
- The results apply to "(C, L)-multiplicatively Lipschitz" kernel functions. The paper provides a formal definition but does not explicitly explain whether popular kernels (e.g., Laplacian, polynomial) satisfy this (C, L)-Lipschitz condition. This makes it harder for a general ICLR audience to grasp the scope of the paper's applicability immediately.

**Questions:**

- Could the authors provide a few examples of common machine learning kernels that satisfy the "(C, L)-multiplicatively Lipschitz" definition?
- What are the main technical barriers to extending your approach to kernels that *do not* satisfy the (C, L)-Lipschitz property
- Could the authors comment on the $\alpha^d = O(poly(n))$ assumption for the adversarial-robust result? Is there some way to preprocess a dataset to decrease the aspect ratio?

---

> ### Author Response · Authors · 2025-11-25
>
> We thank the reviewer for the thorough evaluation and recognition of our technical contributions. We have updated the PDF, and the parts we changed are marked in blue. We will address the reviewers’ concerns below.
>
> W2 and Q1: “Examples of kernels satisfying $(C,L)$-multiplicatively Lipschitz”
>
> Our $(C,L)$-multiplicatively Lipschitz property (Definition B.1) covers several important kernels commonly used in machine learning, including piecewise exponential kernels, polynomial functions of distance (with non-negative coefficients), and rational function kernels (see page 4 in [1]). We add these explicit examples in our revision.
>
> Q2: “Technical barriers for non-$(C,L)$-Lipschitz kernels”
>
> This is actually not a technical barrier, this is a conditional lower bound. As shown by [1] Theorem 1.4 in page 5, assume SETH, there exists a sufficiently large $C_L$, such that there does not exist an efficient algorithm to solve this problem.
>
> W1 and Q3: “The $\alpha^d = O(\mathrm{poly}(n))$ assumption”
>
> In the static setting of [1], they need $d = O(\log n)$. Since we build a dynamic algorithm, which is a harder question. It is natural for us to assume $d = O(\log n)$. However, we would like to note that we will also require $\alpha = O(1)$. We leave relaxing the assumptions on $\alpha$ as a future direction.
>
> We believe these clarifications address the reviewer's concerns!
>
> ### Reference
> [1] Josh Alman, Timothy Chu,  Aaron Schild and Zhao Song. “Algorithms and hardness for linear algebra on geometric graphs”. FOCS 2020.

---

> > ### Comment · Reviewer_rC5G · 2025-11-28
> >
> > Thank you for the detailed response! I’ve read your clarification, and I am happy to maintain my positive assessment.

---

### Official Review · Reviewer_TZMZ · 2025-10-29

**Soundness:** 4
**Presentation:** 3
**Contribution:** 4
**Rating:** 8
**Confidence:** 4

**Summary:**

This paper studied dynamic algorithms for spectral sparsifiers of geometric graphs with $(C, L)$-Lipschitz kernel distance functions. The problem setting is as follows: we have $n$ data points in $\mathbb{R}^{d}$, and the distances between the points satisfy some smoothness conditions (the $(C, L)$-Lipschitz property mentioned in the paper). At every time, an adversary comes to move a point $x_i$ to another position in $\mathbb{R}^{d}$. Our goal is to maintain a spectral sparsifier, i.e., a subset of edges that forms a subgraph $H\subseteq G$, such that the value of $x^{T} L_{H}x$ is close to $x^{T} L_{G}x$, where $L_{H}$ and $L_{G}$ are the graph Laplacians of $H$ and $G$, respectively.

The main result of this paper is a dynamic algorithm that maintains a $(1+\varepsilon)$-approximation for the spectral sparsifier in $n^{1+o(1)}$ pre-processing time and $n^{o(1)}$ amortized update time. Furthermore, when $\Delta^{d}$ is bounded by some polynomial, the algorithm is able to apply some union bound on a special data structure to argue adversarial robustness of their algorithm. Finally, the algorithm is possible to be generalized to approximate matrix multiplications and Laplacian system solvers, which are example applications of spectral sparsifiers.

**Main techniques.** The main techniques of the paper are based on the well-separated pair decomposition ($s$-WSPD) defined in Callahan and Kosaraju [STOC’92]. At a high level, the decomposition finds partitions of vertices such that the shortest distance is at least $s$ times the diameter of the partitions. Therefore, we will only need to preserve the distances between the partitions to obtain good approximations. For the pre-processing steps, we only need dimensionality reduction + random sampling for such a data structure. For dynamic updates, we need to take advantage of the $(C, L)$-Lipschitz property and an algorithm such that each point $x$ only appears in at most $s^d$ WSPD pairs. The algorithm could be made adversarially robust by using a robust distance estimation data structure as long as the aspect ratio and the dimensions are bounded (for the $\varepsilon$-net argument to work).

**Strengths:**

I’m supportive of the paper. The paper studied a well-motivated problem with many downstream applications in numerical linear algebra, data processing, and machine learning. The paper also spent some passages discussing these applications. The intuitions for the techniques are clearly given in the paper, and I could follow most of their ideas. The writing of the paper demonstrates a great breadth of knowledge, and this is the first dynamic algorithm for spectral sparsifier, as far as I know. Therefore, I believe the paper is a solid acceptance.

**Weaknesses:**

I do not see major flaws in the paper. One potential criticism is that the paper contains no experiments; however, I do not think it is an issue for a paper with this amount and quality of results. There are some typos and lower-order clarity issues in the paper (see questions), but I think most of them could be easily fixed.

**Questions:**

Some of the questions are asked in the weakness section. A few additional questions and comments:
- Your title suggested that the algorithm works for kernel graphs. However, I think the algorithm works for general graphs with $(C, L)$-Lipschitz property, right?
- In definition 1.2, why do you need to define a graph here? I assume $L_G$ is an arbitrary matrix instead of the graph Laplacian? If not, then what’s the difference between definitions 1.2 and 1.3?
- If you start with an empty dataset, and get updates for each data point insertion/deletion, and let $n$ be the maximum number of data points we could ever have. In this setting, can your algorithm still get $n^{o(1)}$ amortized update time without any pre-processing time?
- What is the dependency on $\varepsilon$ for your update time? There must be some trade-off, no? I think you’re ignoring some log factors or treating $\varepsilon$ as a constant.
- Did you make the assumption that the total number of updates is $poly(n)$? You probably need that for union bounding over the events. Did you put such a statement/discussion in the paper?
- Line 170/171: I think $\ell_2$ norm is a well-established term, and ‘entry-wise $\ell_2$ norm’ sounds a bit strange.
- Line 201-202: we can compute this partition efficiently *with by*  -> with or by
- Line 217: The notion of the ultra-low dimension JL lemma was mentioned without any explanation or intuition given. I guess the technique is possible to apply due to the problem and distance functions in your setting (in general subspace embedding, $\Omega(\log{n})$ should be necessary). Can you give more explanations about this?
- Line 359: Is there an issue with the notation? Do you mean $d(U^2_1+U^2_2 + \cdots + U^2_k)$?

---

> ### Author Response · Authors · 2025-11-25
>
> We sincerely appreciate your positive comment and your time! We have updated the PDF, and the parts we changed are marked in blue. Let me clarify the following questions:
>
> Q1: “$(C,L)$-Lipschitz property”
>
> You are correct. Our algorithms only work for $(C, L)$-Lipschitz graphs.
>
> Q2: “Difference between Definition 1.2 and 1.3”
>
> $L_G$ is actually a Laplacian matrix for graph $G$.
> Definition 1.2 states the sketching task, that for Laplacian matrix $L_G$ and $x \in \mathbb{R}^n$, we want to maintain a low dimensional sketch of $L_Gx$, i.e. $\\| b - L_Gx\\|_2 \leq \epsilon \cdot \\| L_G \\|\_F \cdot \\|x\\|_2$.
>
> Definition 1.3 states that for Laplacian matrix $L_G$ and $b \in \mathbb{R}^n$, we want to maintain a low dimensional sketch of $L_G^\dagger b$, i.e. $\\|\widetilde{z} - L_G^\dagger b\\|_2 \leq \epsilon \cdot \\|L_G^\dagger\\|\_F \cdot \\|b\\|_2$.
> We apologize for the chaotic notation we used, and we changed it in our revised version on page 3 line 113.
>
> Q3: “Initialize with empty dataset”
>
> In essence, our algorithm is to manipulate dynamic WSPD data structure due to Lemma B.18, our main runtime bottleneck is determined by this. Our dynamic data structure essentially relies on the compressed quad tree data structure due to [5]. Thus the insertion and deletion time is $n^{o(1)}$. Therefore, if we initialize with zero points, it does not affect the $n^{o(1)}$ update time.
>
> Q4: “Dependency on $\epsilon$“
>
> According to Lemma E.5, the running time of initialization is $O(ndk + \epsilon^{-2} n^{1+O(L/k)}2^{O(k)} \log n \log \alpha)$, where the second term depends on $\epsilon^{-2}$.
>
> Q5: “Assumptions on updates”
>
> We do not assume the total number of updates is $poly(n)$.
> Our results contain two algorithms. The first algorithm is non-adversarial, the second one is adversarial.
>
> For the non-adversarial setting (Theorem 2.1),  we essentially have initialization time $t_{\rm init}$, and update time $t_{\rm update}$. We need to find the parameter $K$, such that $K \cdot t_{\rm update} = t_{\rm init}$. If the data structure is updated over $K$ times, we need to restart the data structure by re-initializing it. This ensures the amortized update time remains $n^{o(1)}$. This is formally presented in Lemma E.11 and Corollary E.13.
>
> Such a restarting idea is very standard in the optimization literature with using data-structure. For example in linear programming [1], [2] and [3] (see Appendix line 24 in Algorithm 9 in Section E)
>
> For the adversarial algorithm (Theorem 2.2), we don’t need to restart because we use net argument to handle all the points in the net. (page 8 line 385).
>
> Such an idea is commonly used in the Machine Learning community, for example [4].
>
> Thus we can handle infinitely long updates.
>
> Q6: “Typo in entry-wise”
>
> Thanks for finding this typo. We deleted “entry-wise” in the revised version.
>
> Q7: “Typo”
>
> Thanks for finding this typo. We fixed this in the revision.
>
> Q8: “Notion of Ultra JL Lemma”
>
> Standard subspace embedding is extension of JL with net argument. Here, we are using ultra-JL, a type of dimension reduction that can go beyond $\log(n)$ dimension. However this work is at the expense of getting dimension down to $k$ by paying dilation $n^{O(1/k)}$ and contraction is $1$. This is not $(1\pm\epsilon)$-guarantee. That’s the major difference.
>
> According to Lemma B.6 on page 16:
> For $k = o(\log n)$, with high probability at least $1-1/\mathrm{poly}(n)$ the maximum distortion in pairwise distance obtained from projecting $n$ points into $k$ dimensions (with appropriate scaling) is at most $n^{O(1/k)}$, e.g.,
> $$\\| x - y \\|_2 \leq \\| f(x) - f(y) \\|_2 \leq n^{O(1/k)} \cdot \\| x - y \\|_2$$
> where $f$ is the projection from $\mathbb{R}^d$ to $\mathbb{R}^k$.
>
> Q9: “Typo in $U$ and $V$”
>
> You are correct. We fixed this in the revision.
>
> Again, thanks for your insightful comments!
>
> ### References
> [1] Michael B. Cohen, Yin Tat Lee, and Zhao Song. "Solving linear programs in the current matrix multiplication time." Journal of the ACM (JACM) 68.1 (2021): 1-39.
>
> [2] Yin Tat Lee, Zhao Song, and Qiuyi Zhang. "Solving empirical risk minimization in the current matrix multiplication time." COLT 2019.
>
> [3] Zhao Zhao, and Zheng Yu. "Oblivious sketching-based central path method for linear programming." ICML 2021. Link of Appendix https://proceedings.mlr.press/v139/song21e/song21e-supp.pdf
>
> [4] Yeshwanth Cherapanamjeri, and Jelani Nelson. "On adaptive distance estimation." NeurIPS 2020. https://arxiv.org/abs/2010.11252
>
> [5] Sariel Har-Peled. Geometric approximation algorithms. No. 173. American Mathematical Soc., 2011.

---

### Official Review · Reviewer_oZ3V · 2025-10-31

**Soundness:** 4
**Presentation:** 3
**Contribution:** 3
**Rating:** 4
**Confidence:** 4

**Summary:**

This paper studies the dynamic maintenance of spectral sparsifiers for geometric graphs. Concretely, given an n-vertex geometric graph G (read a complete graph where each edge weight is defined according to some kernel function among its endpoints) associated with a set of points P lying on a d-dimension Euclidean space, the goal is to dynamically maintain a spectral sparsifier H of G under insertions/deletions of points to P. The main result is a data-structure that supports point updates in n^{o(1)} time, while ensuring that the size of H is n^{1+o(1)} at any time. The paper presents a randomized algorithm and discusses extensions that support adaptive adversarial updates. There are also results involving the maintenance of vector-matrix multiplication using random sketches as well as solutions to Laplacian systems of the underlying geometric graph. This review will mainly focus on the dynamic spectral sparsifier contribution.

One obvious approach when thinking about this problem is simply using the dynamic spectral sparsifier algorithm [ADK+16] that handles edge updates to the geometric graph in poly-logarithmic time. However, a single point update might trigger \Omega(n) changes in the complete, geometric graph, which would in turn lead to a linear-time update time guarantee in the worst case. To overcome this issue, the paper leverages the construction of a static, sublinear algorithm for computing geometric spectral sparsifiers [ACSS20].

The algorithm consists of several building blocks. In the first step, it applies a JL transform on the point set P to bring down the dimension of the space, which results in the point set Q. It then proceeds to build a compressed quadtree T of Q and a well-separated pair decomposition on top of T. For each pair in this decomposition, it randomly samples edges from the complete bipartite graph induced by the pair, and the union over all sampled edges forms the final spectral sparsifier of the geometric graph.

The main technical contribution of this work is to make each of these building blocks dynamic. For some of these blocks, dynamic algorithms existed prior to this work, e.g., maintaining a well-separated pair decomposition is a classical result in computational geometry. The paper shows how to adapt this for the problem at hand. To the best of my judgment, it seems that their main contribution in the dynamic maintenance is making sure that they update the uniform samples for the bi-cliques efficiently, and this is also the running time bottleneck of their data structure.

**Strengths:**

* Given the importance of geometric graphs and their application in data-driven applications, the problem studied here is natural and very well motivated. The results on the update time are also competitive.

**Weaknesses:**

* The question is not central to classic dynamic data structures, but perhaps relevant in dynamic ML applications

**Questions:**

n/a

---

> ### Author Response · Authors · 2025-11-25
>
> We thank the reviewer for the thorough evaluation and recognition of our technical contributions. We have updated the PDF, and the parts we changed are marked in blue. We address the concern below:
>
> W1: "Relevance with classic dynamic data structures and dynamic machine learning”
>
> The motivation is many ML applications that require dynamic updates to the kernel, and this inspires us to study this problem. It has applications including kernel methods, dynamic spectral clustering and GNNs.
>
> Kernel Methods: kernel matrices naturally arise in modern machine learning and optimization tasks, from Kernel PCA and ridge regression [1,2,3,4], to Gaussian-process regression [5], federated learning [6], and the 'state-space model' (SSM) in deep learning [7,8].
>
> Dynamic Spectral Clustering: dynamically maintaining a spectral clustering [9] of the kernel matrix is also a fundamental ML application. Recently, [10,11] show the importance of spectral methods. Our dynamic methods could potentially allow the online updating.
>
> Graph Neural Networks (GNNs): Geometric graphs and spectral methods support modern GNN expressivity [12]. Dynamic maintenance could accelerate the applications with evolving graphs (e.g., molecular modeling [13], recommendation systems [14]).
>
> We believe these connections show our work’s relevance with classic dynamic data structures as well as dynamic ML.
>
> Thanks again for your feedback!
>
> ### References:
> [1] Ahmed Alaoui, and Michael W. Mahoney. "Fast randomized kernel ridge regression with statistical guarantees." NeurIPS 2015.
>
> [2] Haim Avron, Kenneth L. Clarkson, and David P. Woodruff. "Sharper bounds for regularized data fitting." In Approximation, Randomization, and Combinatorial Optimization. Algorithms and Techniques (APPROX/RANDOM 2017)
>
> [3] Haim Avron, Michael Kapralov, Cameron Musco, Christopher Musco, Ameya Velingker, and Amir Zandieh. "Random Fourier features for kernel ridge regression: Approximation bounds and statistical guarantees." ICML 2017.
>
> [4] Jason D. Lee, Ruoqi Shen, Zhao Song, Mengdi Wang and Zheng Yu. "Generalized leverage score sampling for neural networks." NeurIPS 2020.
>
> [5] Carl Edward Rasmussen, and Hannes Nickisch. "Gaussian processes for machine learning (GPML) toolbox." The Journal of Machine Learning Research 11 (2010): 3011-3015.
>
> [6] Jakub Konečný, H. Brendan McMahan, Felix X. Yu, Peter Richtárik, Ananda Theertha Suresh, and Dave Bacon. "Federated learning: Strategies for improving communication efficiency." NIPS Workshop on Private Multi-Party Machine Learning 2016.
>
> [7] Albert Gu, Karan Goel, and Christopher Ré. "Efficiently modeling long sequences with structured state spaces." ICLR 2022.
>
> [8] Albert Gu, Isys Johnson, Karan Goel, Khaled Saab, Tri Dao, Atri Rudra, and Christopher Ré. "Combining recurrent, convolutional, and continuous-time models with linear state space layers." NeurIPS 2021.
>
> [9] Andrew Ng, Michael I. Jordan, and Yair Weiss. "On spectral clustering: Analysis and an algorithm." NeurIPS 2001.
>
> [10] Uri Shaham, Kelly P. Stanton, Henry Li, Boaz Nadler, Ronen Basri, and Yuval Kluger. "Spectralnet: Spectral clustering using deep neural networks." ICLR 2018.
>
> [11] Jicong Fan, Yiheng Tu, Zhao Zhang, Mingbo Zhao, and Haijun Zhang. "A simple approach to automated spectral clustering." NeurIPS 2022.
>
> [12] Derek Lim, Joshua David Robinson, Lingxiao Zhao, Tess E. Smidt, Suvrit Sra, Haggai Maron, and Stefanie Jegelka.  "Sign and basis invariant networks for spectral graph representation learning." ICLR 2023
>
> [13] Tian Xie, Arthur France-Lanord, Yanming Wang, Yang Shao-Horn, and Jeffrey C. Grossman. "Graph dynamical networks for unsupervised learning of atomic scale dynamics in materials." Nature communications 10.1 (2019): 2667.
>
> [14] Shiwen Wu, Fei Sun, Wentao Zhang, Xu Xie, and Bin Cui. "Graph neural networks in recommender systems: a survey." ACM Computing Surveys 55.5 (2022): 1-37.

---

### Official Review · Reviewer_oqKw · 2025-11-06

**Soundness:** 3
**Presentation:** 3
**Contribution:** 3
**Rating:** 8
**Confidence:** 3

**Summary:**

The paper studies geometric graph sparsification under the updates that changes a point in the set of input points. They give a sparsification algorithm whose update time is $n^{o(1)}$ and initialization time is $n^{1+o(1)}$. They also give a robust verison of their algorithm under some assumptions.

**Strengths:**

This is in my understanding the first algorithm that provides sparisification of geometric graphs. Using standard techniques, they also extend it to Laplacian solver (though that part I did not find that interesting).

The writing of the paper is good and I could not any weakness or suggestions for the authors to improve their paper. I found it easy to follow, which is a good sign.

I would say that I did not read the proof, but looking at the theorem statement, nothing came out that sounds it is incorrect. I hope the authors have done proper verification of their analysis.

**Weaknesses:**

I did not find any.

**Questions:**

One small quetion: On line 458, you meantion a small caveat: Why is it a problem that x is $n$-dimensional. If I understand correctly, $L_H$ is an n times n matrix, just with few entries. So, if x is sparse, that allows faster multiplication, but its random projection can make it dense and if you are projecting to low-dimensional space, I do not see how matrix-vector multiplication can be done!

---

> ### Author Response · Authors · 2025-11-25
>
> Thanks for your positive comments for our work! We have updated the PDF, and the parts we changed are marked in blue. This is an excellent question! Let me clarify the question.
>
> Q1: “Why do we need sketching?”
>
> Although $L_H$ is sparse, its pseudoinverse $L_H^\dagger$ has no guarantee to be sparse (bottom of page 113 of [1]), and thus the solution $x = L_H^\dagger b$ is a $n$-dimensional vector not necessary sparse. Thus we maintain a sketch of $x$ to allow more efficient dynamic updates.
> Moreover, Laplacian is a M-matrix, all the entries of the inverse of the Laplacian matrix are non-negative. Thus, there is no guarantee that the inverse is sparse [2].
> We updated the explanations in the footnote of page 9.
>
> Again, thanks for your appreciation for our work.
>
> ### Reference
> [1] Yousef Saad. Iterative methods for sparse linear systems. Society for Industrial and Applied Mathematics, 2003. https://www-users.cse.umn.edu/~saad/IterMethBook_2ndEd.pdf.
>
> [2]Samuel I. Daitch, and Daniel A. Spielman. "A nearly-linear time algorithm for approximately solving linear systems in a symmetric m-matrix." Schloss Dagstuhl–Leibniz-Zentrum für Informatik, 2009. https://drops.dagstuhl.de/entities/document/10.4230/DagSemProc.09061.3

---

### Note · Program_Chairs · 2026-02-23
**Submission Desk Rejected by Program Chairs**

The Program Chairs are issuing a desk reject because an author who made foundational contributions to the paper was not credited as an author.
This action does not reflect a judgment on the technical quality, scientific merit, or results of the work. The reviewers originally recommended the paper for acceptance based on its technical contributions.
The Program Chairs recommend that the authors update the authorship information on public pre-print servers and submit the complete work to a future venue to ensure all contributors receive appropriate credit in the final archival record.

---

### Meta-Review · Area_Chair_MC5h · 2026-01-04

**Summary:**

The authors study dynamic linear algebra problems for the graph Laplacian of a kernel-induced graph, under point updates: matrix-vector, solving, and sparsification. The main technical contribution is dynamizing several steps in prior work, e.g., WSPD and associated sampling tasks, which allows for a subpolynomial update time under oblivious queries, and under adaptive with more assumptions.

The problem is natural and all the authors appreciated the final result as well as the technical contributions, so I think this meets the bar for acceptance. I will not recommend a higher rating because the applications for an ML audience are a bit far removed, and I'd like them to be more concrete; despite the theoretical nature of the paper, it would be benefitted substantially by (even a toy example of) a realistic problem where a subset of the authors' techniques can be evaluated empirically.

**Reviewer Concerns:**

The authors clarified several of the technical points and typos brought up by the reviewers. They also added some additional motivation for the problem they study in ML applications. I request that the authors make the promised edits to the final version of the paper.

**Reviewer Scores:**

I believe the reviewer scores are likely to remain relatively similar after considering feedback.

---

### Decision · Program_Chairs · 2026-01-26

Accept (Poster)